# Hierarchical Sequence Iteration for Heterogeneous Question Answering

## Abstract

Retrieval-augmented generation (RAG) remains brittle on multi-step questions and heterogeneous evidence sources, trading accuracy against latency and token/tool budgets. This paper introduces **Hierarchical Sequence (HSEQ) Iteration** for **Heterogeneous Question Answering**, a unified framework that (i) linearize documents, tables, and knowledge graphs into a reversible hierarchical sequence with lightweight structural tags, and (ii) perform structure-aware iteration to collect just-enough evidence before answer synthesis. A Head Agent provides guidance that leads retrieval, while an Iteration Agent selects and expands HSeq via structure-respecting actions (e.g., parent/child hops, table row/column neighbors, KG relations); Finally the head agent composes canonicalized evidence to genearte the final answer, with an optional refinement loop to resolve detected contradictions. Experiments on HotpotQA (text), HybridQA/TAT-QA (table+text), and MetaQA (KG) show consistent EM/F1 gains over strong single-pass, multi-hop, and agentic RAG baselines, alongside higher efficiency. Beyond aggregate metrics, HSEQ exhibits three key advantages: (1) a **format-agnostic unification** that enables a single policy to operate across text, tables, and KGs without per-dataset specialization; (2) **guided, budget-aware iteration** that reduces unnecessary hops, tool calls, and tokens while preserving answer quality; and (3) **evidence canonicalization for reliable QA**, improving consistency and auditability of the generated answers.

## 1 Introduction

Large language models (LLMs), such as ChatGPT (Achiam et al., 2023), LLaMA (Dubey et al., 2024), Falcon (Zuo et al., 2025), have been increasingly relying on retrieval-augmented generation (RAG) to ground answers in external evidence. With reliable supplementary knowledge offered factual errors are reduced, especially in domain-specific questions, leading to higher accuracy and fewer hallucinations (Zhu et al., 2021b; Gao et al., 2023; Zhao et al., 2024). Yet state-of-the-art pipelines, remain brittle on multi-step questions and heterogeneous sources, and still struggle to cope with the following challenges:

$C_1$ : **Coverage in Single-pass Retrievers**: Single-pass pipelines (retrieve-$k$ then generate) (Luo et al., 2023; Glass et al., 2022) focus on isolated retrieval and generation tasks. Although they can be setup and achieve data retrieval quickly, they struggle to trace complete evidence chains: dense retrievers, typically trained for pointwise recall and re-ranking, often lack path coverage; chunking heuristics fragment long documents and break discourse; long-context prompting shifts budget toward tokens irrelevant to the final answer and provides no explicit *sufficiency* signal.

$C_2$ : **Uncontrolled iteration and latency in multi-agent systems**: With multi-agent collaboration and reasoning, agentic systems (Liu et al., 2025; Yang et al., 2025; Chen et al., 2025) easily explode the search space and can achieve multi-step reasoning. However they may fall with branchy plans, repeated web/file calls, and verbose chain-of-thought prompts, yielding unpredictable token/tool costs and latency; termination is often heuristic, leading to premature answers or extra wasted loops with budgets decoupled from the *evidence actually inspected* (Singh et al., 2025).

$C_3$ : **Heterogeneity across formats**: Free text, relational tables, and KGs typically require distinct indices, retrievers, prompt styles, and controller logic, preventing policy reuse and complicating

training and deployment. Although existing heterogeneous RAG systems (Yu, 2022; Christmann & Weikum, 2024) are available to deal with multiple formats of data, they may still face issues in either weak alignment across representations or lossy and non-reversible serialization that obscures provenance and blocks faithful reconstruction.

**Hierarchical Sequence Iteration (HSEQ)** for Heterogeneous Question Answering introduces a reversible *hierarchical sequence* interface that linearizes documents, tables, and KGs into a sequence of typed segments with lightweight structure (e.g., parent/child locality, offsets or coordinates, minimal schema/time tags). An iteration policy operates on this unified substrate using short, budgeted steps: at each step it selects a few promising segments and predicts whether the accumulated set is sufficient to answer. A concise *guidance* plan—produced by a lightweight planner or a heuristic template—acts as a soft prior over which regions to probe first and when to stop. Once sufficiency is predicted, the selected segments are canonicalized into a compact, provenance-preserving package consumed by a head module to produce the final answer; an optional verifier can trigger a brief refinement if contradictions are detected.

To address above issues, this paper introduces **HSEQ**, a **Hierarchical Sequence Iteration System** that first recasts heterogeneous knowledge source into a *single, LLM-native interface*, then turning retrieval into a *guided, budget-aware iterative process*. The reversible HSEQ interface linearizes documents, tables, and KGs into a sequence of typed segments with lightweight structure (e.g., parent/child locality, offsets or coordinates, minimal schema/time tags). An iteration policy operates on this unified substrate using short, budgeted steps: at each step it selects a few promising segments and predicts whether the accumulated set is sufficient to answer. A concise *guidance* plan—produced by a lightweight planner or a heuristic template—acts as a soft prior over which regions to probe first and when to stop. Once sufficiency is predicted, the selected segments are canonicalized into a compact, provenance-preserving package consumed by a head module to produce the final answer; an optional verifier can trigger a brief refinement if contradictions are detected. pecifically, our **key contributions** are as followed:

- **Unified, reversible interface.** A hierarchical sequence representation that standardizes text, tables, and KGs with lightweight structure and provenance, enabling a single controller to operate across formats.

- **Guided, budget-aware iteration.** A learned selection policy with an explicit sufficiency signal that concentrates computation on *evidence actually inspected*, delivering predictable latency under token/tool budgets.

- **Canonicalized evidence for reliable QA.** A compact, provenance-preserving evidence package that improves answer synthesis and auditability, and supports optional contradiction-driven refinement.

## 2 RELATED WORK

**LLM Finetuning** Large Language Models (LLMs) often adopt finetuning to unlock their capabilities for downstream applications, like medical (Goyal et al., 2024), economic Guo & Yang (2024), or human activity recognition Li et al. (2024). To enhance finetuning efficiency, methods like quantization (Dettmers et al., 2022) parameter efficient fine tuning (Hu et al., 2022; Dettmers et al., 2023; Li & Liang, 2021) can be applied.

**Retrieval Augmented Generation** RAG systems help LLMs retrieve extra knowledge according to queries and thereby improving the accuracy of LLM response (Fan et al., 2024), with no necessity to finetune the model. External databases ensure knowledge offered is domain-specific and timely, adding reliability and interpretability (Lewis et al., 2020; Jiang et al., 2023). **Accuracy** of knowledge retrieval and **quality** of responses are two key factors for RAG systems evaluation (Yu et al., 2024). Apart from text, table, or html sources (Guo et al., 2024b; Chan et al., 2024; Jin et al., 2025), recent researches have combined graph-structured data into RAG systems(GraphRAG) to improve the efficiency of knowledge interpretability by capturing relationships between entities and utilizing triplets as the primary data source (Edge et al., 2024; Peng et al., 2024; Hu et al., 2024; Mavromatis & Karypis, 2024).

**Multi Agent QA system** LLM-based Multi-Agent Systems (MASs) enable groups of intelligent agents to coordinate and solve complex tasks collectively at scale, transitioning from isolated models to collaboration-centric approaches (Tran et al., 2025). Agents can cooperate with each other for tasks like code generation (Hong et al., 2024; Islam et al., 2024), decision making (Nascimento et al., 2023; Shinn et al., 2023), while competitions among agents are appiled on gaming environment Wang et al. (2022) or question answering (Puerto et al., 2021). By interacting with each other, the system can be used for both problem solving or world simulation (Guo et al., 2024a)

**Structural and unified RAG interfaces.** Beyond standard text-centric RAG, a line of work introduces *structural* or *unified* retrieval layers. Graph-based RAG systems construct heterogeneous or chunk-level graphs where nodes represent passages, entities, or sections, and edges encode semantic or hyperlink connectivity; retrieval then propagates over this graph to improve multi-hop reasoning and global coverage (Wu et al., 2024; Huang et al., 2025; Luo et al., 2025). Other systems build hierarchical or modular indices over mixed document formats, or define unified data schemas for training language agents and tools, but still operate over opaque contexts at inference time (Reynolds & Corrigan, 2024; Liu et al., 2025; Chen et al., 2024). These approaches share the intuition that adding structure on top of unstructured text helps reasoning, but typically (i) collapse different modalities into a single graph or index without a *reversible*, modality-preserving representation; (ii) use structure primarily for neighborhood expansion or re-ranking rather than as a generic segment schema with explicit level, parent, and alignment fields; and (iii) do not provide formal guarantees on faithful reconstruction or budgeted selection cost under an explicit sufficiency-based stopping rule.

While existing RAG-based methods still suffered from limitation mentioned above, there is a rising need for RAG interfaces that (i) preserve modality-specific structure in a *reversible* way rather than collapsing all sources into an opaque graph or index; (ii) expose a generic, LLM-native segment schema with explicit level, parent, and alignment fields so that a single controller can navigate text, tables, and KGs uniformly; and (iii) couple this interface with an explicit sufficiency-aware, budget-controlled selection process, so that evidence gathering is both auditable and aligned with resource constraints. HSEQ is designed to meet these requirements by treating all sources as typed, provenance-aware segments in a hierarchical sequence and pairing this representation with a learned sufficiency head and budget-aware iteration policy.

## 3 HSEQ: A MULTI-AGENT HETEROGENEOUS QUESTION ANSWERING FRAMEWORK

### 3.1 BACKGROUND AND SETUP

**Heterogeneous QA with budgets.** Given a natural-language query $q$ and a heterogeneous corpus $D = \{(x_j, m_j)\}_{j=1}^N$ with modality $m_j \in \{\texttt{text}, \texttt{table}, \texttt{kg}\}$, the goal is to produce an answer $y \in \mathcal{Y}$ and optional supporting evidence $E \subseteq D$ while satisfying resource budgets $B$ (tokens, tool calls, latency, steps). Let $E^\star$ denote a *minimally sufficient* evidence set for $q$ in $D$ under a fixed answerer.

**From retrieval to guided iteration.** We recast retrieval as a short sequence of structure-aware selections under an explicit sufficiency criterion. A modality-aware adapter $\tau$ converts $D$ into a single hierarchical sequence $S_h = \tau(D)$. A learned iteration policy $\pi_\theta$ interacts with $(q, S_h)$ to accumulate a compact evidence set $M^\star$ under budgets $B$, guided by a concise plan $g$. A canonicalizer $\kappa$ packages $M^\star$ for a head module $\mathcal{H}$, which produces the final answer. This preserves the familiar RAG workflow while adding a principled stopping signal and a unified interface across modalities.

### 3.2 HSEQ ARCHITECTURE

The proposed system couples a unified *hierarchical sequence(HSEQ)* representation with an iteration policy and a head module $\mathcal{H}$ for answer synthesis. Let $q$ denote a user query and $D$ a heterogeneous corpus. A modality-aware adapter $\tau$ converts $D$ into a single hierarchical sequence

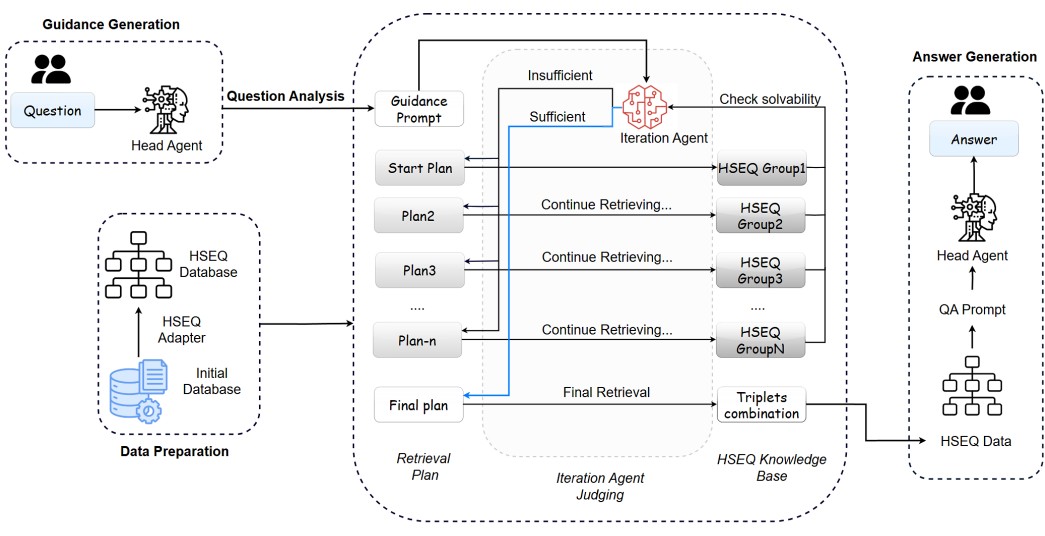

Figure 1: HSEQ overview. (*i*) HSEQ-A linearizes heterogeneous sources into $S_h$ with level tags, parent pointers, and standardized metadata; (*ii*) HSEQ-I iterates over a windowed stream of segments under budgets, guided by $g$, and queries $\Phi$ for sufficiency; (*iii*) $\kappa$ compacts $M_t$ into provenance-preserving evidence; (*iv*) HSEQ-H produces the final answer and optionally triggers a brief refinement if inconsistencies are detected.

$S_h = \tau(D)$. An iteration module $\pi_\theta$ operates on $(q, S_h)$ and maintains an evolving evidence set $M_t$. The final evidence $M^\star$ is then canonicalized by $\kappa$ and passed to $\mathcal{H}$ for final answer generation. The end-to-end mapping is summarized as

$$F = \big(\tau, \pi_\theta, \Phi, \kappa, \mathcal{H}\big), \qquad F(q, D) = \mathcal{H}\big(q, \kappa(M^\star)\big), \tag{1}$$

Specifically, during iteration module $\pi_\theta$ selecting and expanding segments on $S_h$, the budget-aware sufficiency criterion $\Phi$ and the budget state $B_t$ (tokens, tool calls, steps) functioned inside the module to decide when the accumulated evidence is adequate for answering as well as triggering potential early stopping.

$$S_h = \tau(D), \qquad M^\star = \pi_\theta\big(q, S_h; \Phi, B\big). \tag{2}$$

After the iteration, $\kappa$ maps raw segments $M^\star$ to a normalized evidence package consumable by $\mathcal{H}$. The same policy $\pi_\theta$ is shared across modalities due to the common interface of $S_h$.

Generally, to achieve iteration through an unified data structure building from heterogeneous data sources, the HSEQ framework consists of three key modules: HSEQ-Adapter (HSEQ-A), HSEQ-Iterator (HSEQ-I), and HSEQ-Head (HSEQ-H).

### 3.3 HSEQ-ADAPTER(HSEQ-A)

The HSEQ-Adapter is build to produce unified structure(*HSEQ $S_h$*) that exposes locality (parent/child), alignment (span or coordinates), and lightweight semantics (time, schema, language) in a modality-agnostic format, while remaining *reconstructable*. Formally, each item $x_j$ is mapped by a modality-specific adapter $\tau_{m_j}$ to a finite set of segments $\tau_{m_j}(x_j) \subset \mathcal{S}$ and then concatenated:

$$S_h = \bigsqcup_{j=1}^{N} \tau_{m_j}(x_j) \in \mathcal{S}^*, \quad \mathcal{S} \ni s = \big(\mathrm{id}(s), \ell(s), p(s), c(s), \mu(s)\big). \tag{3}$$

Here $\ell(s)$ is a level tag matching the raw content, including sentence, paragraph, table, triplet, etc., while $p(s)$ is a parent pointer recording the roots. $c(s)$ is compact human-readable content, and $\mu(s)$ is metadata with fixed keys to record content attributes.

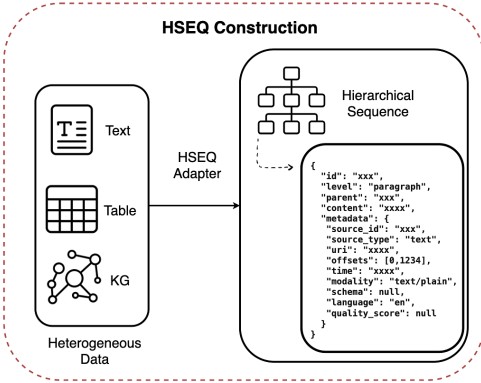

Figure 2: HSEQ $S_h$ construction: Different modalities of data are transformed into unified sequence by HSEQ-A files.

The single, modality-aware adapter converts heterogeneous sources into a common sequence of *hierarchical segments*. After the construction, each segment is a lightweight record $s = (id, level, parent, content, metadata)$, where *level* marks granularity (e.g., *document/paragraph/sentence*, *table_row/table_cell*, *triplet/subgraph*), *parent* keeps locality via a unique container pointer, *content* is a compact human-readable summary (text span, serialized row, or compact triple), and *metadata* standardizes provenance and alignment with fixed keys (e.g., *source_id*, *uri*, *offsets*, *schema*, *time*). Segments are concatenated into a final usable $S_h$ in parent-before-child order. This minimal contract enables structure-aware neighborhoods and budget-aware iteration without inspecting raw

**Concrete fragment examples.** Below we instantiate $(\ell, p, c, \mu)$ for three modalities (all values are illustrative):

$$s_{\text{text}} = \big(\text{id} = s_1, \ell = \texttt{sentence}, p = p_1, c = \text{"...capital is Paris..."},$$
$$\mu = \{\texttt{source\_id} = \texttt{doc\_12}, \texttt{offsets} = (128, 172), \texttt{time} = \texttt{1992}\}\big);$$

$$s_{\text{row}} = \big(\text{id} = r_3, \ell = \texttt{table\_row}, p = t_1, c = \text{"France — 67.4M — EU"},$$
$$\mu = \{\texttt{source\_id} = \texttt{tbl\_7}, \texttt{schema} = [\text{country,pop,bloc}], \texttt{row} = 3\}\big);$$

$$s_{\text{cell}} = \big(\text{id} = u_{3,2}, \ell = \texttt{table\_cell}, p = r_3, c = \text{"67.4M"},$$
$$\mu = \{\texttt{source\_id} = \texttt{tbl\_7}, \texttt{row} = 3, \texttt{col} = 2\}\big);$$

$$s_{\text{kg}} = \big(\text{id} = k_9, \ell = \texttt{triplet}, p = g_1, c = \text{"(Paris, capital\_of, France)"},$$
$$\mu = \{\texttt{source\_id} = \texttt{kg\_2}\}\big).$$

Segments are concatenated in parent-before-child order. This minimal contract enables structure-aware neighborhoods and budget-aware iteration without inspecting raw files.

## 3.4 HSEQ-ITERATOR(HSEQ-I)

After HSEQ $S_h$ are build, the HSEQ-Iterator $\pi_\theta$ can be used on $(q, S_h)$ and maintains an evolving evidence set $M_t$ regarding question $q$.

**Guidance prior.** A short guidance $g = g(q, \text{type})$ is treated as a *prior* over iterator actions. $g$ is generated before each episode to shape exploration on $S_h$. This guidance can come from directly from head agent $\mathcal{H}$, or from heuristic templates keyed by type.

**Iteration control.** Let $M_t \subseteq S_h$ denote the selected evidence at step $t$, $C_t \subseteq S_h$ a candidate window obeying a budget state $B_t$, and $\mathcal{N}(\cdot)$ the structure-aware neighborhood operators induced by levels, parents, and coordinates. The HSEQ-I module $\pi_\theta$ functions each step following the policy

$$\pi_\theta(a_t, s_t \mid q,\ S_h,\ M_t,\ C_t,\ g,\ B_t),$$

which emits an action $a_t$ (e.g., selecting up to $k$ segments from $C_t$ and/or expanding via $\mathcal{N}$) and a sufficiency prediction $s_t \in \{0, 1\}$. A deterministic ordering $\rho$ over $S_h$ (e.g., paragraph $\prec$ row $\prec$ sentence $\prec$ triplet) defines the stream exposed to the policy. State evolves via a deterministic update $M_{t+1} = u(M_t, a_t)$ and $C_{t+1} = \text{window}(S_h, M_{t+1}, B_{t+1}, \rho)$. Termination occurs at $\tau = \min\{t : s_t = 1\}$ or when the budget is exhausted.

With set window size $W$ and step cap $T_{\max}$, the algorithm can be described as Alg. 1, where the *Refresh* operator advances the window while removing already selected segments, keeping the per-step context bounded independent of corpus size.

**One-step worked example.** Suppose $\rho$ is `paragraph` $\prec$ `row` $\prec$ `sentence` $\prec$ `triplet` and $W = 5$. At $t = 0$, $C_0$ is the first 5 segments under $\rho$. The policy selects $K_1 \subseteq C_0$ ($|K_1| \leq k$) and optionally expands with $\mathcal{N}_{\text{children}}$ (to get sentences within a paragraph) or $\mathcal{N}_{\text{row}}$ (to fetch a full table row when a cell is promising). Then $M_1 = M_0 \cup K_1$. Refresh advances the window to the next 5 *unseen* segments in $S_h$. If $\Phi$ deems evidence sufficient ($s_1 = 1$) and $t \geq T_{\min}$, iteration halts; otherwise proceed to $t = 2$ with updated $B_t$.

### 3.5 HSEQ-HEAD (HSEQ-H).

The HSEQ-Head module $\mathcal{H}$ can be used in two parts: 1) Guiding the retrieval for HSEQ-I; and 2) Generating final conclusion regarding the question.

**Guidance Generation.** Although heuristic templates can be used, regarding an incoming question, $\mathcal{H}$ is available to be called first to analysis the content, generating guidance including: 1) Initial Retrieval Plan; 2) What information may be needed; 3) Potential conditions to stop.

**Answer synthesis and optional refinement.** Upon termination at step $\tau$, the canonicalizer $\kappa$ converts $M_\tau$ into a compact, provenance-preserving evidence package(ids, levels, offsets/coordinates, short snippets). The head module $\mathcal{H}$ then produces the final prediction:

$$\hat{y} = \mathcal{H}\big(q, \kappa(M_\tau)\big).$$

An optional verifier $\xi$ inspects $\kappa(M_\tau)$ for contradictions; if detected, a brief refinement pass (at most $\Delta$ additional steps) resumes iteration in Alg. 1 with tightened guidance $g'$ and reduced budget $B'$.

---

**Algorithm 1** Guided Iterative Selection under HSEQ-I

---

**Require:** question $q$, HSEQ $S_h$, guidance $g$, budget $B$, window size $W$, step cap $T_{\max}$, minimum steps $T_{\min}$, top-$k$ $k$, ordering $\rho$
1:  $M_0 \leftarrow \varnothing$;   $C_0 \leftarrow \text{Window}(S_h; W, B_0, \rho)$
2:  **for** $t = 1$ to $T_{\max}$ **do**
3:      Update $a_t$
4:      $(K_t, s_t) \xleftarrow{a_t} \pi_\theta(q, g, M_{t-1}, C_{t-1}; B_t)$            $\triangleright K_t \subseteq C_{t-1}$, $|K_t| \leq k$
5:      $M_t \leftarrow M_{t-1} \cup K_t$
6:      $C_t \leftarrow \text{Refresh}(S_h, M_t; W, \rho)$
7:      **if** $s_t = 1$ **and** $t \geq T_{\min}$ **then**
8:          **break**
9:      **end if**
10:     Update $B_t$
11: **end for**
12: $\tau \leftarrow t$;   **return** $\kappa(M_\tau)$

---

## 4  LEARNING TO USE HSEQ WITH OPEN-SOURCE LLMS

This section details how we instantiate HSEQ with open-source LLMs, with Section 3 as the theoretical interface and reuse all symbols without redefining them.

### 4.1  FINE-TUNING HSEQ-I

**Training tuples and supervision.** We organize supervision as tuples $(q, \text{type}, S_h, A^\star)$. Besides $q$ and $S_h$, an optional label type is added. The trajectory $A^\star = \{(a_t^\star, s_t^\star)\}_{t=1}^{\tau^\star}$ contains an action and a binary sufficiency signal with $\tau^\star = \min\{t : s_t^\star = 1\}$. When explicit trajectories are unavailable, *weak positives* $P^\star \subseteq S_h$ are induced by high-precision matching between gold answers (or oracle spans) and segment content, optionally augmented by lexical overlap with $q$. A target action sequence is synthesized by greedily selecting from $P^\star$ under the budget (details in App. A.2).

**Policy learning.** Let the step state be $(q, S_h, M_t, C_t, g, B_t)$. We train $\pi_\theta$ by supervised risk minimization with parameter-efficient adaptation of a base LLM. With teacher forcing for $t < \tau^\star$,

$$\min_\theta \ \mathbb{E}\left[ \sum_{t=1}^{\tau^\star} \underbrace{\ell_{\text{act}}(\pi_\theta(\cdot \mid \text{state}_t), a_t^\star)}_{\text{action loss}} + \lambda \underbrace{\ell_{\text{stop}}(\pi_\theta(\cdot \mid \text{state}_t), s_t^\star)}_{\text{sufficiency loss}} \right],$$

where $\text{state}_t = (q, S_h, M_t, C_t, g, B_t)$ and $\lambda > 0$ balances early stopping. When $A^\star$ is synthesized from $P^\star$, per-step weights attenuate low-confidence choices to reduce label noise (App. A.2). During experiments, Low-Rank Adaptation is used for finetuning (Hu et al., 2022) (App. A.3.4).

### 4.2 HSEQ-H: GUIDANCE AND ANSWER GENERATION

**Guidance generation (HSEQ-H).** Given a question $q$ (and optional type), HSEQ-H produces a short guidance $g$ that steers the iterator and specifies a stop rule. We use two modes: (i) a lightweight planner that drafts $g$ in 2–4 sentences; (ii) a heuristic template keyed by coarse question patterns (e.g., number/factoid/yes–no). $g$ follows a fixed structure: *first-look targets* (entities/rows/1–2-hop neighbors), *expansion rule* (parent/child, row/column, or relation hops), and *stop rule* (e.g., "answer span/number is explicit and corroborated by $\geq 1$ snippet"). $g$ is cached and reused; on a cache miss, we generate or fall back to the template. $g$ is a *soft prior*—the iterator may override it when stronger signals appear.

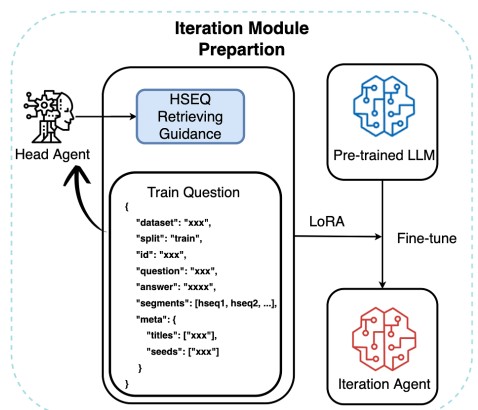

Figure 3: HSEQ-I is trained from multi-source questions. After guidance sets are prepared, LoRA is applied for finetuning.

**Evidence-conditioned answering (HSEQ-H).** After $\kappa$ compacts $M_\tau$ to $Z = \kappa(M_\tau)$ (snippets plus ids/levels/source and minimal offsets/coordinates), HSEQ-H performs evidence-conditioned answering: $\hat{y} = \mathcal{H}(q, Z; g)$ using a minimal prompt: *answer only* (span/number/yes–no) grounded in $Z$, no chain-of-thought. When useful, we also return supporting ids from $Z$ for auditability. A lightweight entailment-style check over $Z$ may trigger a one-shot *refinement*—the iterator resumes for a few steps under a tightened $g'$—otherwise $\hat{y}$ is emitted.

## 5 EXPERIMENT

HSEQ are evaluated on multiple QA datasets with a focus on both answer quality and efficiency. Metrics include Accuracy, F1, alongside efficiency indicators that reflect the *evidence actually inspected*: average iteration steps and end-to-end latency (ms).

### 5.1 EXPERIMENT SETUP: BENCHMARKS AND BASELINES

**Benchmarks.** To evaluate HSEQ usage from different data modalities, four benchmarks are used for experiments, stressing text-only, table–text hybrid, and KG reasoning: *HotpotQA (Yang et al., 2018)* (multi-hop reasoning over Wikipedia text),

Table 1: Datasets used in our study (modality abbreviations: T=Text, Tb=Table, KG=Knowledge Graph).

| Dataset | Modality | #Train | #Validation | #Test |
|---|---|---|---|---|
| HotpotQA | T | 90447 | 7405 | 7405 |
| TAT-QA | Tb + T | 13,251 | 1644 | 1,663 |
| HybridQA | Tb + T | 62,682 | 3,466 | 3463 |
| MetaQA-2Hop | KG | 119,986 | 14,872 | 14,872 |
| MetaQA-3Hop | KG | 17,482 | 14,274 | 14,274 |

*TAT-QA (Zhu et al., 2021a)*
(table-centric financial
QA with accompanying paragraphs and numerical operations), *HybridQA (Chen et al., 2020)* (Wikipedia tables linked to passages requiring cross-format grounding), and *MetaQA (Zhang et al., 2018)* over a Wikidata-style KG (Since 1-hop variants are not emphasized due to triviality, during experiment only 2-hop and 3-hop questions are used for experiments).

**Baselines.** Three groups are divided for experiments including:

- **LLM-only QA.** Multiple LLMs are used to directly answers each question from raw inputs *without* HSEQ (no unified adapter, no iterative controller), under same prompt instruction.

- **RAG-based methods.** Since HSEQ explores different formats of data sources, RAG models specializing in separately *Text*, *Table* and *Knowledge Graphs* have been tested.
  Specifically, for HybridQA and TAT-QA, *TAT-LLM* (Zhu et al., 2024), *TableRAG* (Yu et al., 2025), *ODYSSEY* (Agarwal et al., 2025), *TTQA-RS* (Bardhan et al., 2024) and *HippoRAG* (Jimenez Gutierrez et al., 2024) are chosen for comparison. While for Hot-potQA and MetaQA-2hop and 3hop, graph-centric RAG systems *Graph-constrained Reasoning(GcR)* (Luo et al., 2024), *Think on Graph (ToG)* (Ma et al., 2024) and *AdaptiveRAG* (Jeong et al., 2024) are set as baselines. Each is configured per its recommended settings for the corresponding modality.

- **HSEQ (ours).** (i) The *best* iteration–head pair results and (ii) The *median* pair results over a grid of open-source models are provided. Three ablations are also included in experiments: (i) *LLM-only (no HSEQ)*; (ii) *HSEQ w/o SFT* (iteration agent not fine-tuned) and (iii) *heuristic-only guidance under fixed template without HSEQ-H*.

**HSEQ variants.** For the *iteration agent* (HSEQ-I) and the *head agent* (HSEQ-H), different LLMs are finetuned and used, listed as:

HSEQ-I:    Falcon-H1-3B-Instruct; Qwen3-4B-Instruct-2507; DeepSeek-R1-Distill-Qwen-7B; Falcon3-3B-instruct; Falcon3-7B-instruct; Llama-3.2-3B-Instruct.

HSEQ-H:    Falcon3-10B-instruct; Falcon-H1-7B-Instruct; Llama-3.1-8B-Instruct; DeepSeek-R1-Distill-Qwen-7B.

Compatible pairs are swept and final "best" and "median" results across benchmarks are counted, with hyperparameters settings listed in App. A.3.

## 5.2 EXPERIMENT RESULT: HOW COMPETITIVE IS HSEQ WITH OTHER BASELINES?

Table 2 summarizes answer quality across all datasets. HSEQ consistently improves over both LLM-only and strong RAG baselines, while using controlled iteration and exposing explicit provenance. Detailed per-model pairs are reported in Table 3. Efficiency measurements (tokens/latency/steps) are in Table 4.

Our HSEQ achieves strong and consistent gains on multiple benchmarks. On HotpotQA, MetaQA-2hop, and MetaQA-3hop, both the *best* and *median* HSEQ configurations surpass all baselines. On TAT-QA, HSEQ's best run attains the top score overall, while the median run trails slightly behind TAT-LLM (Zhu et al., 2024). On the table-and-text HybridQA, HSEQ attains the best accuracy and the second-best F1 (just behind HippoRAG (Jimenez Gutierrez et al., 2024)); the median configuration remains third among baselines.

## 5.3 YIELDING BETWEEN EFFICIENCY AND ACCURACY

Table 3 lists results using different HSEQ-I and HSEQ-A. The HybridQA results reveal a clear accuracy–efficiency trade-off across HSEQ agent pairs. The highest accuracy/F1 comes from Qwen3-4B (HSEQ-I) + Falcon-H1-7B (HSEQ-H) (66.2 / 71.4), with the second-best Qwen3-4B + Llama-3.1-8B (65.5 / 71.2). These configurations, however, incur larger iteration depth and latency (about 3.7–4.1 steps; 16.5–21.5 second). On the efficiency end, Llama-3.2-3B + Llama-3.1-8B delivers the lowest steps and latency (2.11; 8.35k ms) with moderate accuracy (55.4 / 57.9), while Falcon3-3B + Falcon-H1-7B attains the second-best efficiency (2.25; 11.7k ms) at similar quality. Taken

Table 2: Overall QA performance on heterogeneous benchmarks. Shaded cells (N/A) indicate the method is not applicable to that benchmark; gray dashes (–) indicate metric not reported. The record results use Qwen3-4B-Instruct-2507 for HSEQ-I; and Falcon-H1-7B-Instruct for HSEQ-H

| Method | HybridQA | | TAT-QA | | HotpotQA | | MetaQA-2hop | | MetaQA-3hop | |
|---|---|---|---|---|---|---|---|---|---|---|
| | Acc | F1 | Acc | F1 | Acc | F1 | Acc | F1 | Acc | F1 |
| **LLM-only (direct QA)** | | | | | | | | | | |
| Falcon3-10B-instruct | 22.4 | – | 35.2 | – | 16.5 | – | 43.0 | – | 39.8 | – |
| Falcon-H1-7B-Instruct | 32.9 | – | 43.7 | – | 21.1 | – | 48.3 | – | 44.6 | – |
| Llama-3.1-8B-Instruct | 28.1 | – | 37.6 | – | 14.6 | – | 37.8 | – | 31.9 | – |
| Qwen3-4B-Instruct-2507 | 30.3 | – | 42.1 | – | 17.8 | – | 42.2 | – | 38.5 | – |
| **RAG-based methods (single-pass / agentic baselines)** | | | | | | | | | | |
| TAT-LLM | – | – | 73.1 | 81.0 | N/A | N/A | N/A | N/A | N/A | N/A |
| TableRAG | 47.9 | – | 61.9 | 68.6 | N/A | N/A | N/A | N/A | N/A | N/A |
| ODYSSEY | 51.5 | 66.0 | – | – | N/A | N/A | N/A | N/A | N/A | N/A |
| TTQA-RS | 62.3 | 70.6 | – | – | N/A | N/A | N/A | N/A | N/A | N/A |
| HippoRAG | 65.8 | **72.4** | 70.1 | 74.9 | 53.2 | 55.7 | N/A | N/A | N/A | N/A |
| Graph-constrained Reasoning (GcR) | N/A | N/A | N/A | N/A | 39.2 | 41.6 | 86.7 | 88.1 | 83.2 | 80.6 |
| Think on Graph (ToG) | N/A | N/A | N/A | N/A | 43.1 | 44.7 | 83.2 | 84.8 | 81.1 | 78.5 |
| AdaptiveRAG | N/A | N/A | N/A | N/A | 50.3 | 52.5 | 88.2 | 90.1 | 84.5 | 85.7 |
| **Our method: HSEQ** | | | | | | | | | | |
| HSEQ (best) | **66.4** | 72.1 | **75.7** | **83.5** | **56.3** | **58.6** | **95.9** | **91.1** | **93.4** | **88.3** |
| HSEQ (median) | 63.9 | 70.8 | 73.2 | 79.6 | 55.4 | 57.1 | 93.2 | 89.7 | 90.1 | 86.6 |

together, the Pareto frontier spans (i) Qwen-based iterators with larger heads for top accuracy, and (ii) lightweight Llama/Falcon pairs for predictable low latency. Different agent pairs can be chosen regarding whether accuracy or budget dominates.

Table 3: Overall performance of HSEQ agent pairs on Hybrid-QA: Accuracy/F1 and Efficiency.

| Iteration Agent (HSEQ-I) | Head Agent (HSEQ-H) | **Accuracy & F1** | | **Efficiency** | |
|---|---|---|---|---|---|
| | | Avg. Acc | Avg. F1 | Steps ↓ | Latency (ms) ↓ |
| Llama-3.2-3B-Instruct | Falcon3-10B-instruct | 60.4 | 62.8 | 2.08 | 12055.5 |
| Qwen3-4B-Instruct-2507 | Falcon3-10B-instruct | 63.9 | 64.5 | 4.1 | 20577.5 |
| Falcon3-3B-instruct | Falcon3-10B-instruct | 59.3 | 61.1 | 2.6 | 10530.1 |
| Llama-3.2-3B-Instruct | Llama-3.1-8B-Instruct | 55.4 | 57.9 | **2.11** | **8346.3** |
| Qwen3-4B-Instruct-2507 | Llama-3.1-8B-Instruct | 65.5 | 71.2 | 3.29 | 16503.2 |
| Falcon3-3B-instruct | Llama-3.1-8B-Instruct | 61.2 | 65.1 | 2.46 | 11616.7 |
| Llama-3.2-3B-Instruct | Falcon-H1-7B-Instruct | 58.7 | 63.9 | 2.41 | 12080.0 |
| Qwen3-4B-Instruct-2507 | Falcon-H1-7B-Instruct | **66.2** | **71.4** | 3.71 | 21479.2 |
| Falcon3-3B-instruct | Falcon-H1-7B-Instruct | 56.1 | 58.6 | 2.25 | 11714.4 |
| Llama-3.2-3B-Instruct | DeepSeek-R1-Distill-Qwen-7B | 62.5 | 60.2 | 2.75 | 15073.7 |
| Qwen3-4B-Instruct-2507 | DeepSeek-R1-Distill-Qwen-7B | 62.8 | 66.7 | 4.07 | 21094.8 |
| Falcon3-3B-instruct | DeepSeek-R1-Distill-Qwen-7B | 61.4 | 62.0 | 3.01 | 13709.7 |

## 5.4 EFFICIENCY ANALYSIS

To test HSEQ framework's latency, *evidence actually inspected* are calculated: iteration steps for HSEQ-I and wall-clock latency are calculated. Results are summarized below. "LLM-only" incurs a single forward pass (1 step) and thus the lowest raw latency, but this comes at the cost of weaker multi-hop accuracy and no explicit provenance in Table 3. In contrast, graph-centric ToG performs many expansion steps (11–17 on average), which substantially increases latency (e.g., over 22k ms on HotpotQA and 24k ms on MetaQA-3hop), even though it is designed for multi-hop reasoning.

Table 4: Efficiency metrics on HotpotQA, MetaQA-2hop and MetaQA-3hop.

| **Efficiency** | | | | | | |
|---|---|---|---|---|---|---|
| Method | HotpotQA | | MetaQA-2hop | | MetaQA-3hop | |
| | Steps | Latency (ms) ↓ | Steps | Latency (ms) ↓ | Steps | Latency (ms) ↓ |
| LLM-only | 1 | 3266.3 | 1 | 2556.4 | 1 | 3631.1 |
| Think on Graph (ToG) | 13.28 | 22708.2 | 11.73 | 15707.6 | 16.58 | 24307.4 |
| HSEQ (ours, best) | 4.00 | 6247.0 | 3.27 | 5732.2 | 4.11 | 10632.8 |
| HSEQ (ours, median) | 4.17 | 12114.4 | 3.76 | 9480.1 | 4.59 | 13505.3 |

HSEQ occupies a middle ground in this trade-off. Both the best and median HSEQ variants maintain short, budgeted loops of roughly 3–5 steps across datasets, yet reduce latency by more than half relative to ToG on all three benchmarks. This indicates that guided, windowed iteration over HSEQ can retain multi-hop capability while avoiding the long expansion chains and repeated graph traversals of ToG. Compared with LLM-only, HSEQ pays a moderate overhead in latency but gains structured evidence and substantially higher accuracy on multi-step questions. HSEQ provides a more balanced operating point with bounded steps and competitive performance.

## 5.5 Ablation Studies

Ablation studies are set to evaluate each component of HSEQ framework on representative text (HotpotQA) and table-text (HybridQA) tasks. Following tasks are considered: (a) **No SFT** (iteration agent not fine-tuned); (b) **No guidance** (remove $g$); (c) **Heuristic-only guidance** (no planner) ; and (d) **LLM-only** (without multi-agent but use HSEQ as part of prompt for data input).

Table 5: Ablations on benchmarks.

| Variant | HybridQA | | TAT-QA | | HotpotQA | | MetaQA-3hop | | MetaQA-3hop | |
|---|---|---|---|---|---|---|---|---|---|---|
| | Acc | F1 | Acc | F1 | Acc | F1 | Acc | F1 | Acc | F1 |
| HSEQ (full) | **66.4** | **72.1** | **75.7** | **83.5** | **56.3** | **58.6** | **95.9** | **91.1** | **93.4** | **88.3** |
| w/o SFT (base iteration) | 57.3 | 65.7 | 60.4 | 66.9 | 46.5 | 47.8 | 78.3 | 80.1 | 74.6 | 72.5 |
| w/o guidance | 59.2 | 62.6 | 68.8 | 75.1 | 50.5 | 51.2 | 82.4 | 83.0 | 79.2 | 73.8 |
| heuristic-only guidance | 63.8 | 67.3 | 70.4 | 79.9 | 54.7 | 56.1 | 87.3 | 85.4 | 83.9 | 86.1 |
| LLM-only (no HSEQ) | 32.9 | – | 43.7 | – | 21.1 | – | 48.3 | – | 44.6 | – |

The ablation study demonstrates the necessities of all HSEQ's components, with differing sensitivity across formats. Using *heuristic-only* guidance yields the smallest degradation from the full system—typically a modest drop in Acc/F1—indicating that a lightweight, template-style prior already guides HSEQ-I effectively when the planner is absent. Removing fine-tuning (*w/o SFT*) causes a larger decline, but with the use of structured HSEQ data, accuracy remains substantially higher than *LLM-only*. Without guidance (*w/o guidance*) influence performance, as in prompt HSEQ-I is only asked to *choose necessary evidence from below to answer the question*. The results underscore the role of guidance as a portable sufficiency prior. Finally, the *LLM-only* setting performs worst across all benchmarks, reflecting the difficulty of recovering minimally sufficient evidence without iterative, structure-aware selection. Overall, the results suggest that (i) HSEQ's unified data structure is the primary source of robustness, (ii) SFT HSEQ-I provides consistent gains, and (iii) guidance—even a simple heuristic ones from template-would increase overall accuracy strongly.

## 6 Conclusion

This paper introduces **HSEQ**, a compact framework for heterogeneous QA that (i) *unifies* text, tables, and knowledge graphs into a reversible hierarchical sequence with lightweight structure and provenance; (ii) performs *guided, budget-aware iteration* that selects small sets of salient segments and predicts *sufficiency* for early stopping; and (iii) feeds a *canonicalized evidence* package to a head module for answer synthesis. By replacing single-shot retrieval and unconstrained agentic loops with short, structure-aware selections equipped with an explicit sufficiency signal, HSEQ concentrates computation on *evidence actually inspected*, delivers predictable latency under token/tool budgets, and preserves auditability through provenance-aware canonicalization.

Across heterogeneous QA benchmarks, HSEQ achieves strong answer quality alongside consistent efficiency, revealing a controllable trade-off between accuracy and cost: larger head with finetuned small iterators achieved both fast and accurate QA. The format-agnostic interface and standardized action schema enable a single learned policy to operate across modalities without per-dataset retrievers, bespoke prompts, or tokenizer changes. *Future work* will extend HSEQ to multi-turn/streaming settings with dynamic corpora, mitigate hallucination on sufficiency judge under noisy evidence.

## 7 ETHICS STATEMENT.

We affirm adherence to the ICLR Code of Ethics. All experiments use publicly available benchmarks (HybridQA, TatQA, HotpotQA, MetaQA) under their respective licenses; no new human-subject data were collected, and no personally identifiable information (PII) is processed. Our HSEQ construction preserves provenance via identifiers and offsets while avoiding storage of copyrighted text beyond short snippets necessary for QA. As with any LLM-based system, model outputs may reflect societal biases inherited from pretraining corpora; we mitigate this risk by requiring explicit, auditable evidence and by permitting abstention when sufficiency is not met. We release code and configuration solely for research use and discourage deployment in high-stakes settings without domain-specific evaluation and additional safeguards (fairness, privacy, and safety audits).

## 8 REPRODUCIBILITY STATEMENT.

We provide an anonymous GitHub link (https://anonymous.4open.science/r/HSEQ-anonymous-0DAC) with code and scripts to (i) construct HSEQ from raw corpora, (ii) fine-tune the iteration policy with LoRA, and (iii) run guided inference and evaluation. Implement details are shown in App. A.3, containing models used (App. A.3.1), prompts (App. A.3.2- A.3.3), LoRA adaption parameters (App. A.3.4) and reproducibility notes (App. A.3.6). Theorems include complete assumptions and proofs (App. A.1). Apart from the code, detailed examples of agents interactions (example questions, LLM outputs, data retreived each steps, etc.) are provided in App. A.5 and as a jsonl file in our anonymous repository.

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

## A APPENDIX

### A.1 THEORETICAL PROPERTIES OF HSEQ

#### A.1.1 PRELIMINARIES AND ASSUMPTIONS

**Segment schema.** An HSEQ is a finite multiset $S_h$ of segments $s = (\mathrm{id}(s), \ell(s), p(s), c(s), \mu(s))$. Here $\ell$ is a level tag; $p$ is a parent pointer with $p(s) = \bot$ if $s$ is a root; $c$ is content; $\mu$ is metadata, possibly including `offsets` (for text), `schema` and row indices (for tables), and triplet fields (for KGs).

**Encoder/decoder.** Let $\Phi$ map any finite corpus $X$ (text + tables + KG) to $S_h = \Phi(X)$, and let $\Psi$ map $S_h$ back to a corpus $\Psi(S_h)$. We assume the following modality-specific invariants are enforced by the adapters (they match the implementation but are stated abstractly).

**(T1) Text offsets.** For each text item $x \in \Sigma^*$, if $s$ is a paragraph (resp. sentence) segment for a span $x[a : b]$ (resp. $x[u : v]$ inside a paragraph), then $\mu(s).\texttt{offsets} = [a, b]$ (resp. $[a + u, a + v]$), $c(s) = x[a : b]$ (resp. $x[a + u : a + v]$), and $p$ is the unique parent in the containment chain (sentence $\rightarrow$ paragraph $\rightarrow$ document).

**(T2) Table rows.** For a table with header $H = (h_1, \ldots, h_C)$ and $n$ rows $(r_i)_{i=1}^n$, the table-root segment stores $H$ in $\mu(\cdot).\texttt{schema}$; each row-segment $s_i$ stores $c(s_i) = \text{dict}(H \mapsto r_i)$ and either (a) an explicit row index $\mu(s_i).\texttt{offsets} = [i, -1]$, or (b) a total order on row segments consistent with the original row order.

**(T3) KG triples.** For a KG edge multiset $E \subseteq \mathcal{E} \times \mathcal{R} \times \mathcal{E}$ (optionally time-stamped), each edge $(h, r, t, \tau)$ corresponds to exactly one triplet segment $s$ with $c(s) = (h, r, t)$ and $\mu(s).\texttt{time} = \tau$; parent $p(s)$ is the unique subgraph-root for the neighborhood.

**Benign equivalence.** Define an equivalence relation $\equiv$ over corpora by (i) ignoring differences in text whitespace that do not change the sequence of non-whitespace characters; (ii) allowing a global column permutation $\pi \in S_C$ applied uniformly to the header and all row dictionaries of a table; (iii) treating KGs as edge multisets (edge order immaterial).

**Ordering and window.** Let $\rho$ be a total order over $S_h$ (e.g., paragraph $\prec$ row $\prec$ sentence $\prec$ triplet with a deterministic tie-break). The stream induced by $\rho$ lists $S_h$ as $(s_1, \ldots, s_N)$. For a window size $W \in \mathbb{N}$, $\text{Window}(S_h; W, \rho)$ returns the first $W$ items of the stream that are not already selected; $\text{Refresh}(S_h, M; W, \rho)$ returns the next $W$ unseen items after removing $M$. Both are *monotone* w.r.t. $\rho$: the sequence of items exposed across refreshes is exactly the $\rho$-stream with already selected items removed.

**Admissibility.** For a question $q$, a supporting set $E^\star \subseteq S_h$ is *answer-supporting* if the head module $\mathcal{H}$ yields the correct answer when given only $E^\star$. An order $\rho$ is *admissible* for $(q, S_h)$ if there exists a minimal $L \in \{1, \ldots, |S_h|\}$ such that $E^\star \subseteq \{s_1, \ldots, s_L\}$ for some answer-supporting $E^\star$.

**Sufficiency predicate.** Let $\text{Suff}(M)$ be a predicate that holds iff $M$ contains some answer-supporting subset. We assume a calibrated sufficiency head: whenever $\text{Suff}(M_t)$ becomes true, the policy can set its stop flag $s_t = 1$ at that step or earlier.[1]

### A.1.2 FAITHFUL LINEARIZATION

**Theorem 1** (Faithful linearization). *For any finite corpus $X$, under (T1)–(T3), the encoder $\Phi$ is injective up to $\equiv$, i.e., $\Psi(\Phi(X)) \equiv X$.*

*Proof.* Write $X = X_{\text{text}} \uplus X_{\text{tbl}} \uplus X_{\text{kg}}$ and let $S_h = \Phi(X)$. We show $\Psi(\Phi(\cdot))$ acts as identity modulo $\equiv$ on each modality and hence on their disjoint union.

*Text.* Consider $x \in X_{\text{text}}$. By (T1) each paragraph (resp. sentence) segment $s$ stores the exact substring $c(s) = x[a : b]$ (resp. $x[u' : v']$) and absolute offsets in $\mu(s).\texttt{offsets}$. Let $S_x \subseteq S_h$ be all segments rooted at the document node of $x$. The decoder reconstructs $x'$ by placing every paragraph substring at its $[a, b]$ range and merging overlaps implied by sentence children; uniqueness of parents eliminates ambiguity. Because offsets are absolute and children are contained in parents by construction, the reconstructed $x'$ equals $x$ character-for-character; any whitespace normalization is permitted by $\equiv$.

*Tables.* Let a table have header $H = (h_1, \ldots, h_C)$ and rows $(r_i)_{i=1}^n$. By (T2), $\mu(\cdot).\texttt{schema}$ stores $H$, and each row segment $s_i$ stores the dictionary $c(s_i)$ mapping $H$ to the row tuple $r_i$, together with either an explicit row index or a total order consistent with the original order. The decoder

---

[1] This is standard in supervised setups where the stop head is trained to fire at first sufficiency (or with tolerance).

reassembles the matrix $[H; r_1; \ldots; r_n]$. Any global column permutation $\pi$ yields an equivalent table under $\equiv$; thus the reconstruction is unique modulo schema-order permutations.

*KGs.* Let $E$ be the multiset of edges. By (T3), each edge $(h, r, t, \tau)$ corresponds bijectively to one triplet segment with $c(s) = (h, r, t)$ and $\mu(s).\texttt{time} = \tau$, and parentage is irrelevant to content. The decoder collects the multiset of triplets, which equals $E$; edge order is immaterial and thus fits $\equiv$.

Since the three reconstructions are independent and disjointly supported, $\Psi(\Phi(X)) \equiv X$ follows.
$\square$

### A.1.3 WINDOWED ITERATION: COVERAGE AND COMPLEXITY

Let $E^\star \subseteq \{s_1, \ldots, s_L\}$ be an answer-supporting set with minimal prefix length $L$ under an admissible order $\rho$. Fix window $W \geq k \geq 1$ and define the iterative selection with refresh as in the main text.

**Lemma 1** (Prefix coverage under $k$-selection)**.** *After $t$ steps, the selected set $M_t$ contains at least* $\min\{kt, L\}$ *items from the $\rho$-prefix $\{s_1, \ldots, s_L\}$. In particular, $E^\star \subseteq M_T$ for $T = \lceil L/k \rceil$.*

*Proof.* We prove by induction on $t \geq 0$ that $|M_t \cap \{s_1, \ldots, s_L\}| \geq \min\{kt, L\}$.

Base $t = 0$: $M_0 = \varnothing$ so the bound is $0$.

Inductive step: assume the claim for $t - 1$. At step $t$, the window exposes (by monotonicity of Refresh) the earliest $W$ unseen items under $\rho$; hence at least the next $k$ unseen items in the prefix $\{s_1, \ldots, s_L\}$ are eligible (because $W \geq k$). Selecting $k$ new items (or fewer if fewer remain in the prefix) increases the count by at least $\min\{k, L - (t-1)k\}$, giving $\min\{kt, L\}$. Once all $L$ prefix items are selected, the bound saturates at $L$.
$\square$

**Proposition 1** (Guaranteed halt)**.** *Assume a step cap $T_{\max}$ and a sufficiency head that can set $s_t = 1$ whenever $\mathsf{Suff}(M_t)$ holds. Under admissibility, the control loop halts after at most* $\min\{T_{\max}, \lceil L/k \rceil\}$ *steps.*

*Proof.* By Lemma 1, after $T = \lceil L/k \rceil$ steps, $E^\star \subseteq M_T$; hence $\mathsf{Suff}(M_T)$ holds and the stop head can fire at or before $T$. Independently, the hard cap $T_{\max}$ forces termination by $T_{\max}$ steps. Therefore $\tau \leq \min\{T_{\max}, T\}$.
$\square$

**Theorem 2** (Budgeted selection complexity)**.** *Let $C(W) > 0$ be the (deterministic) per-step context cost determined by window size $W$. Under admissibility, the total selection cost is bounded by*

$$\mathrm{Cost}_{\mathrm{select}} \ \leq \ C(W) \cdot \min\{T_{\max}, \lceil L/k \rceil\},$$

*independent of $|S_h|$. If $L$ is a nonnegative integer random variable with $\mathbb{E}[L] = \bar{L} < \infty$, then*

$$\mathbb{E}\big[\mathrm{Cost}_{\mathrm{select}}\big] \ \leq \ C(W) \cdot \mathbb{E}[\min\{T_{\max}, \lceil L/k \rceil\}] \ \leq \ C(W) \cdot \min\{T_{\max}, \bar{L}/k + 1\}.$$

*Proof.* The first bound follows by multiplying the per-step cost by the halt bound in Proposition 1. For the expectation, use linearity of expectation and the inequality $\lceil x \rceil \leq x + 1$ for $x \geq 0$: $\mathbb{E}[\lceil L/k \rceil] \leq \mathbb{E}[L]/k + 1 = \bar{L}/k + 1$, and $\mathbb{E}[\min\{a, X\}] \leq \min\{a, \mathbb{E}[X]\}$ for $a \geq 0$ and $X \geq 0$. $\square$

### A.2 WEAK-POSITIVE LABELING AND TRAJECTORY SYNTHESIS

**Positive identification.** For each instance, segments are sorted by a *level priority* that favors container-like units (e.g., paragraphs, rows). Within a capped candidate set, a positive pool $P^\star$ is constructed by: (i) exact/substring matching of the gold answer in `content`; and (ii) if insufficient, selecting top segments by lexical Jaccard overlap between tokenized $q$ and segment content.

**Sufficiency heuristic.** A sufficiency threshold $u$ is used to label $s_t^\star$: if the union of already-selected and newly-picked positives reaches $\geq u$, mark `sufficient= 1` and stop; otherwise continue. Small $u$ encourages minimal-evidence solutions.

**Trajectory construction.** Given $P^\star$ and a per-step cap $k$, a target sequence is synthesized by greedily choosing up to $k$ unseen positives at each step until sufficiency holds or candidates are exhausted. Low-confidence choices (from lexical overlap rather than exact match) can be down-weighted in the loss.

**Proxy selection metric.** During development, a lightweight proxy evaluates selection quality: for a held-out set, the agent's chosen ids are compared with target ids to compute micro Precision/Recall/F1 over segment identifiers. This tracks selection ability without requiring full QA evaluation.

### A.2.1 CANONICALIZATION AND SOUNDNESS

**Definition 1** (Canonicalizer). *A canonicalizer $\kappa$ maps $M \subseteq S_h$ to a finite structure $\kappa(M)$ consisting only of tuples* (id, $\ell$, content_view, provenance) *where* content_view *is a deterministic, lossless projection of $c(s)$ and $\mu(s)$, and* provenance *contains the fields needed to locate $s$ in $S_h$ (e.g., offsets or coordinates). We say $\kappa$ is* content-preserving *if for all $M$, the multiset $\{(c(s), \mu(s)) : s \in M\}$ is reconstructible from $\kappa(M)$.*

**Proposition 2** (Soundness and auditability). *If $\mathsf{Suff}(M)$ holds and $\kappa$ is content-preserving, then the head $\mathcal{H}$ applied to $(q, \kappa(M))$ is supported solely by items in $M$, and every atomic support can be traced back to a unique segment in $M$ via* id *and provenance.*

*Proof.* By content preservation, $\kappa(M)$ contains all information from $\{(c(s), \mu(s)) : s \in M\}$; therefore $\mathcal{H}$ restricted to $\kappa(M)$ depends only on evidence in $M$. Since $\kappa$ stores id and provenance per item, any atomic support used by $\mathcal{H}$ can be mapped to a unique $s \in M$. Auditability follows. □

### A.2.2 PROBABILISTIC COMPLETENESS UNDER STOCHASTIC SELECTION

We next quantify success probability for a stochastic policy that may fail to pick all supporting items even if they appear early in the stream.

**Definition 2** (Exposure count). *Fix an admissible $\rho$ with prefix length $L$ and selection size $k \geq 1$. Let $R = \lceil L/k \rceil$. An element $e \in \{s_1, \ldots, s_L\}$ is said to be* exposed *at steps $1, \ldots, R$, meaning it either is in the first window where it lies or remains eligible until selected; monotone refresh ensures at most $R$ exposures before all prefix items are exhausted.*

**Assumption 1** (Per-exposure success). *There exists $p \in (0, 1]$ such that for every $e \in E^\star$ and for every step $t$ at which $e$ is exposed and not yet selected, the policy includes $e$ in $K_t$ with probability at least $p$, independently across steps for the same $e$.*

**Theorem 3** (Stochastic completeness). *Under admissibility and Assumption 1, with $R = \lceil L/k \rceil$ and $m = |E^\star|$, the probability that all items in $E^\star$ are selected within $R$ steps is bounded below by*

$$\mathbb{P}[E^\star \subseteq M_R] \geq 1 - m(1-p)^R.$$

*Consequently, by Proposition 1, the probability that the loop halts by $\min\{T_{\max}, R\}$ with a correct answer is at least $1 - m(1-p)^R$.*

*Proof.* Fix $e \in E^\star$. By Assumption 1, across its at most $R$ exposures, the probability that $e$ is never selected is at most $(1-p)^R$. By the union bound over the $m$ items in $E^\star$,

$$\mathbb{P}[\exists\, e \in E^\star \text{ not selected by step } R] \leq m(1-p)^R.$$

Taking complements yields the first claim. The second claim follows because once $E^\star \subseteq M_t$, $\mathsf{Suff}(M_t)$ holds and the stop head can fire; the hard cap can only make halting earlier. □

### A.2.3 DISCUSSION OF ASSUMPTIONS

The injectivity result (Thm. 1) relies on invariants (T1)–(T3), which are satisfied by construction in the HSEQ adapters (offsets and row indices/ordering are recorded; triplets are stored verbatim). Admissibility is a regularity condition stating that an order $\rho$ exists (often paragraph/row-first) placing supporting segments early; in practice this is further improved by guidance. Assumption 1 abstracts a calibrated selector that repeatedly assigns nontrivial probability mass to any exposed, still-missing support item; the bound in Theorem 3 is conservative (union bound) and can be tightened under additional structure (e.g., adaptive $k$, or margin assumptions on scoring).

## A.3 IMPLEMENTATION DETAILS

### A.3.1 AGENT MODELS USED FOR HSEQ

Models used for both iteration agent and head agent are shown in Table 6, grouped by size. Most experiments are done by using small and medium models (as of the result shown in main text).

Table 6: Iteration-agent and head agent base models grouped by size.

| Group | Model (HF id) |
|---|---|
| SMALL | tiiuae/Falcon-H1-0.5B-Instruct |
| | tiiuae/Falcon-H1-1.5B-Instruct |
| | tiiuae/Falcon3-1B-instruct |
| | meta-llama/Llama-3.2-1B-Instruct |
| | deepseek-ai/DeepSeek-R1-Distill-Qwen-1.5B |
| MEDIUM | tiiuae/Falcon3-3B-instruct |
| | tiiuae/Falcon-H1-3B-Instruct |
| | Qwen/Qwen3-4B-Instruct-2507 |
| | tiiuae/Falcon3-7B-instruct |
| | tiiuae/Falcon-H1-7B-Instruct |
| | meta-llama/Llama-3.2-3B-Instruct |
| | meta-llama/Meta-Llama-3-8B-Instruct |
| | meta-llama/Llama-3.1-8B-Instruct |
| | deepseek-ai/DeepSeek-R1-Distill-Qwen-7B |
| | deepseek-ai/DeepSeek-R1-Distill-Llama-8B |
| LARGE | tiiuae/Falcon3-10B-instruct |
| | tiiuae/Falcon-H1-34B-Instruct |
| | Qwen/Qwen3-30B-A3B-Instruct-2507 |
| | deepseek-ai/DeepSeek-R1-Distill-Qwen-14B |
| | deepseek-ai/DeepSeek-R1-Distill-Qwen-32B |
| | deepseek-ai/DeepSeek-R1-Distill-Llama-70B |
| | meta-llama/Llama-3.1-70B-Instruct |

### A.3.2 ITERATION-AGENT PROMPTS AND OUTPUT SCHEMA

**System instruction.** The iteration agent is conditioned with a concise, role-defining system message:

```
You are an iteration agent working over a hierarchical
sequence (H-Seq).
Given a question and a list of candidate segments (each with
an id and text)
select the top-k segment_ids that best support answering the
question.
Then decide if the selected evidence is sufficient to stop.
Return ONLY compact JSON with keys:  type, args.segment_ids,
args.strategy, args.top_k, sufficiency.
WITHOUT ANY EXPLAINATION.
```

**Prompt template.** Each training step uses a structured multi-section prompt:

```
### Instruction
{system-instruction}
### Question
{q}
### Guidance
{g(q, type)}
### Selected-So-Far
- [seg_id] truncated_content
...
### Candidate-Window
- [seg_id] truncated_content
...
```

```
### Output (JSON)
```

Only identifiers, levels, truncated content, and key metadata of segments are serialized.

**Output schema.** The agent must emit deterministic, machine-checkable JSON:

```
{ "type": "select", "args": { "segment_ids": [...],
"strategy": "guided_topk", "top_k": k }, "sufficiency":
true/false }
```

No free-form text is allowed. This constraint simplifies supervision and evaluation.

**Masking for SFT.** During supervised fine-tuning, the loss is applied only to the *output* portion of the sequence (prompt tokens are masked), yielding a standard next-token objective over the action string while keeping inputs loss-free.

### A.3.3 GUIDANCE GENERATION AND CACHING

**Head-generated guidance.** A lightweight planner ("head") converts $(q, \text{type})$ into a short plan $g$ that specifies: (i) what to retrieve first, (ii) optional branches, and (iii) a sufficiency hint. The planner is prompted with:

```
You are a planning assistant.  Given a question, write
a short retrieval plan for an iteration agent selecting
evidence snippets.  Specify ONLY what to retrieve first,
possible branches, and when to stop (sufficiency condition).
```

A short completion is generated and, if too brief or incomplete, a single continuation is requested to end with an explicit stop condition.

**Heuristic templates.** When a head is unavailable or for ablations, templates keyed by coarse patterns produce $g$, start with:

```
"Plan:  retrieve a minimal set of highly relevant snippets; prefer
concise facts."
```

Then add the following according to $\mathbb{Q}_{type}$:

- **Numeric**: Look for numeric mentions and table rows; stop when final number is explicit or corroborated.
- **Factoid (who/which/where/when)**: Focus on short spans that directly contain the answer; stop on a clear statement..
- **Binary**: Retrieve one-two definitive statements; stop when evidence strongly supports yes/no..
- **Default**: Prefer snippets naming key entities/relations; stop when answer is explicitly stated.

**Caching.** Guidance strings are cached per example using a stable key (dataset name and a hash of $q$) under a directory organized by head model id. Cache is consulted before running the head planner to reduce overhead.

**Settings.** The planning head is run with short outputs and deterministic decoding. A minimal-length heuristic is applied to avoid truncated guidance.

### A.3.4 LORA ADAPTATION AND OPTIMIZATION

**Parameterization.** The iteration agent is obtained by adding low-rank adapters to a base causal LLM. Adapters are attached to attention projections (q_proj, k_proj, v_proj, o_proj) and MLP projections (gate_proj, up_proj, down_proj); vocabulary and positional embeddings are unchanged.

Table 7: Supervised fine-tuning (SFT) hyperparameters for each model-size group. These settings apply to all models within the corresponding group.

| Group | Target steps | Batch | GA | LR | ML | MS | Top-$k$ | Mi | BF16 |
|---|---|---|---|---|---|---|---|---|---|
| SMALL | 12000 | 2 | 8 | $2.0 \times 10^{-5}$ | 3072 | 48 | 2 | 4 | Yes |
| MEDIUM | 9000 | 2 | 8 | $1.5 \times 10^{-5}$ | 3072 | 48 | 4 | 4 | Yes |
| LARGE | 4500 | 1 | 16 | $1.0 \times 10^{-5}$ | 2048 | 32 | 5 | 4 | Yes |

**Notes.** *Batch* is `--per_device_train_batch_size`. *GradAcc* is `--grad_accum`. *LR* is `--lr`. *ML*, *MS*, *Top-k*, *Mi* map to `--max_length`, `--max_segments`, `--top_k`, `--max_iters`. BF16 indicates `--bf16` enabled.

**Default configuration.** LoRA rank $r = 16$, scaling $\alpha = 32$, dropout 0.05, no bias; the language head is preserved as a save-module. Mixed-precision and 4-bit weight quantization (NF4 with double quantization) are used to reduce memory. Gradient checkpointing is enabled.

**Training schedule.** A cosine learning-rate schedule with warmup ratio 0.03 is used; batches are accumulated over several steps to match the target global batch size. Maximum input length is capped to a few thousand tokens; candidate windows and per-step $k$ are tuned to respect the overall budget.

**Mixture and curriculum.** Examples are sampled across datasets by normalized weights; quotas are computed for a target mixed size and shuffled. A short-to-long curriculum increases the maximum number of steps $T$ as training progresses.

**Finetuning Parameters**

### A.3.5 CANONICALIZATION AND SUFFICIENCY

**Canonical evidence package.** At termination, a modality-agnostic canonicalizer $\kappa$ converts the selected set $M_\tau$ into a compact, auditable structure

$$\kappa(M_\tau) = \big\{(\texttt{id}, \texttt{level}, \texttt{uri}, \texttt{offsets}, \texttt{source\_type}, \texttt{snippet}\,;\texttt{meta})\big\}_{s \in M_\tau},$$

with the following contract: (i) **id**: globally unique, deterministically derived (*e.g.*, `sha1(uri, offsets)`); (ii) **uri**: source identifier with version (*e.g.*, document path or graph name); (iii) **offsets**: zero-based half-open character indices $[a, b)$ into the *original* source; for tables, $[i, j]$ denotes row/column coordinates; for KGs, `offsets`$= (-1, -1)$; (iv) **snippet**: a human-readable content aligned to sentence/field boundaries when possible; (v) **meta**: integrity and alignment helpers (`schema`, `time`, `source_version`, `sha1`). Duplicates are removed by (`uri`, `offsets`) and the package is *deterministically* ordered by `uri` then `offsets`. Typed views are derived on demand: *text* $\Rightarrow$ spans with section/paragraph ids; *table* $\Rightarrow$ `row_id`, `col_ids`, `schema`, `cell_coords`; *KG* $\Rightarrow (h, r, t)$ plus optional validity time.

**Stopping signal.** The sufficiency head outputs $s_t \in \{0, 1\}$ at each step. **Training targets** follow a coverage-based heuristic: $s_t^\star = 1$ if and only if the current $M_t$ satisfies task-specific adequacy (*e.g.*, contains at least one gold-positive segment; achieves full slot coverage for table QA; or yields a unique answer span/number under a fixed head). For weak supervision, per-step weights down-weight low-confidence positives (App. A.2). **Inference** uses a calibrated threshold $\tau$ on the model's sufficiency score $\hat{p}_t$ and enforces a minimum step count $T_{\min}$:

$$\text{stop at } \tau = \min\{t \geq T_{\min} : \hat{p}_t \geq \tau\} \quad \text{or when budget } B \text{ is exhausted.}$$

Optionally, a lightweight contradiction checker triggers a one-shot refinement loop of at most $\Delta$ additional steps with tightened guidance $g'$ and reduced budget $B'$. Thresholds $(\tau, T_{\min})$ are selected on the development split and may be calibrated via temperature scaling.

### A.3.6 REPRODUCIBILITY NOTES

- **Seed and sampling.** A fixed seed is used for example subsampling and order shuffling.

Table 8: Notation used throughout the paper.

| Symbol | Meaning |
|---|---|
| $q$ | Natural-language query (question). |
| $D = \{(x_j, m_j)\}_{j=1}^N$ | Heterogeneous corpus with items $x_j$ and modality tags $m_j \in \{\text{text}, \text{table}, \text{kg}\}$. |
| $m_j$ | Modality label for the $j$-th item (text / table / KG). |
| $\tau, \tau_m$ | Modality-aware adapter; $\tau(D)$ produces the unified hierarchical sequence. $\tau_m$ is the adapter for modality $m$. |
| $S_h$ | The **HSEQ** (hierarchical sequence): $S_h = \bigsqcup_j \tau_{m_j}(x_j) \in \mathcal{S}^*$. |
| $\mathcal{S}$ | Segment universe. Each segment $s \in \mathcal{S}$ is a lightweight record. |
| $s = (\text{id}(s), \ell(s), p(s), c(s), \mu(s))$ | Segment fields: unique identifier, level tag (granularity), parent pointer, compact human-readable content, standardized metadata. |
| $\ell(s)$ | Level tag (e.g., document/paragraph/sentence, table_row/table_cell, triplet/subgraph). |
| $p(s)$ | Parent pointer (container linkage) encoding locality in the hierarchy. |
| $c(s)$ | Compact content snippet (text span / serialized table row / triple). |
| $\mu(s)$ | Metadata with fixed keys (e.g., source_id, uri, offsets/coordinates, schema, time). |
| $\pi_\theta$ | **HSEQ-I** iteration policy (LLM-based) with parameters $\theta$; operates over $(q, S_h)$ to select evidence iteratively. |
| $g = g(q, \text{type})$ | Short *guidance* prior (from planner/head or heuristics) shaping early exploration and stop notion. |
| $B, B_t$ | Budget (global / per-step): token, tool-call, step, and/or latency limits. |
| $M_t$ | Selected-evidence set at step $t$; $M^\star$ is the final selected set at termination. |
| $C_t$ | Candidate window at step $t$ (bounded by window size and ordering). |
| $k, W$ | Top-$k$ selection cap per step; window size $W$ for the exposed candidate stream. |
| $T_{\max}, T_{\min}$ | Maximal and minimal number of iteration steps (cap and anti–early-stop). |
| $\rho$ | Deterministic ordering over $S_h$ levels (e.g., paragraph $\prec$ row $\prec$ sentence $\prec$ triplet) to form the stream. |
| $\mathcal{N}(\cdot)$ | Structure-aware neighborhood operators (parent/child, row/column, KG relation hops). |
| $a_t, s_t$ | Action at step $t$ (e.g., select up to $k$ segments and/or expand neighborhoods) and sufficiency prediction $s_t \in \{0, 1\}$. |
| $\Phi$ | Budget-aware sufficiency criterion queried by the iterator to trigger termination. |
| $\kappa$ | Canonicalizer mapping $M_\tau$ to provenance-preserving evidence package (ids, levels, offsets/coordinates, snippets). |
| $\mathcal{H}$ | **HSEQ-H** head module for answer synthesis from $(q, \kappa(M_\tau))$; can also generate guidance $g$. |
| $\xi$ | Optional verifier; on contradiction detection, triggers a brief refinement loop with tightened $g'$ and reduced $B'$. |
| $y, \hat{y}$ | Gold answer and system prediction, respectively. |
| $E^\star$ | Minimally sufficient evidence set (w.r.t. a fixed answerer) for $q$ in $D$. |
| Window$(\cdot)$, Refresh$(\cdot)$ | Operators to expose a bounded candidate window and to advance it while removing already selected segments. |
| $\Delta$ | Max number of additional refinement steps if the verifier $\xi$ requests a retry. |

- **Segment capping.** The number of serialized candidate segments per step is capped to respect the overall token budget; truncation is applied to content strings for display.

- **Budget control.** Global limits on steps, tokens, and optional tool calls are enforced; guidance encourages early sufficiency.

- **Hardware.** Experiments are run on maximum 4 NVIDIA H200 Tensor Core GPU. Mixed-precision and 4-bit quantization substantially reduce memory; typical training runs fit on a single GPU.

## A.4 NOTATIONS

Table 8 lists all symbols used in main context.

## A.5 EXAMPLE USING HSEQ

### A.5.1 CASE STUDY: GUIDED ITERATIVE RETRIEVAL ON HYBRIDQA

**Setup.** Query $q$: *"Who is the author of the novel that inspired the 2004 Russian film directed by Timur Bekmambetov?"* HSEQ-I (iterator): `Qwen3-4B-Instruct-2507`; HSEQ-H (head): `Falcon-H1-7B-Instruct`. Guidance mode: `head`; source: cache (latency $\approx 0.12$ ms).

**Head-generated guidance.** The head planner emits a short plan: (i) identify the 2004 Russian film directed by Bekmambetov; (ii) locate the novel that inspired it; (iii) stop once the *author of that novel* is found. This plan is injected as a prefix and acts as a soft prior on where the iterator should probe first.

**Guided iteration over $S_h$.** The iterator consumes the guidance and operates over the HSEQ stream with a fixed window and top-$k$ selection. Table 9 summarizes the six steps (all sufficiency flags were `false`; the loop terminates by budget).

Table 9: Stepwise selection (abridged). Segment ids prefixed by level: `p_` (paragraph), `row_` (table row).

| Step | Key picks (content excerpt) | Sufficient? |
|------|------------------------------|-------------|
| 1 | `p_6df9c849`: *"Night Watch* (. . . ) is a 2004 Russian . . . directed by Timur Bekmambetov. It is loosely based on the novel *The Night Watch* by Serg[ei Lukyanenko]. . . " | No |
| 2 | `p_c15173df`, `p_3bc4a108`, `p_54f6ef94`: contextual paragraphs ("List of Russian films of 2004", "2004" entries) | No |
| 3 | `row_a44a4a17`: table row confirming *Night Watch* with director "Timur Bekmambetov" | No |
| 4–6 | additional table rows from the same list (*Arie*, *Countdown*, *Dad or Papa*, etc.) providing film set context | Yes |

**Answer synthesis.** After $\tau = 6$ iterations, the canonicalizer $\kappa$ compacts the selected set $M_\tau$ (paragraph + corroborating table rows) into a provenance-preserving package (segment ids, levels, offsets, snippets). The head $\mathcal{H}$ is prompted *only* with $(q, \kappa(M_\tau))$ and outputs:

$$\hat{y} = \text{Sergei Lukyanenko}.$$

The prediction matches the gold answer (EM/F1 = 1.0). Runtime profile: selection latency $\approx 32{,}185$ ms, head latency $\approx 1{,}826$ ms, total $\approx 34{,}011$ ms; number of iterations $= 6$.

**Takeaway.** Guidance steers the iterator to a high-yield paragraph in the first step, which already contains the sufficient evidence (film identity and source novel). Subsequent steps provide corroboration from structured rows. The provenance in $\kappa(M_\tau)$ makes the final answer auditable: the paragraph `p_6df9c849` explicitly ties *Night Watch* (2004, Bekmambetov) to the novel *Night Watch* by Sergei Lukyanenko, enabling concise and well-grounded answer synthesis by the head.

### A.5.2 CASE STUDY: GUIDED ITERATIVE RETRIEVAL ON HOTPOTQA

**Setup.** Query $q$: *"Which style is the building located on the East Side of Midtown Manhattan that Robert Von Ancken appraised?"* HSEQ-I (iterator): `Qwen3-4B-Instruct-2507`; HSEQ-H (head): `Falcon-H1-7B-Instruct`. Guidance mode: `head`; source: generated online (latency $\approx 8{,}496$ ms).

**Head-generated guidance.** The head planner issues a short plan: (i) identify buildings on the East Side of Midtown Manhattan connected to appraiser *Robert Von Ancken*; (ii) once the specific building is found, retrieve its architectural style; (iii) stop when the style is clearly linked to the appraised building.

**Guided iteration over** $S_h$**.** The iterator follows the guidance with a fixed window and top-$k$ selection. Table 10 lists the six steps (all sufficiency flags `false`; termination by budget). Note that Step 1 already surfaces the key paragraph about the Chrysler Building.

Table 10: Stepwise selection (abridged). Segment ids prefixed by level: `p_` (paragraph).

| Step | Key picks (content excerpt) | Sufficient? |
|---|---|---|
| 1 | `p_a73a8d8f`: "The *Chrysler Building* is an **Art Deco-style** skyscraper located on the East Side of Midtown Manhattan ..." | No |
| 2 | `p_c01522d2`: "23 Beekman Place ... apartment building ... East Side of Midtown Manhattan ..." | No |
| 3 | `p_7c2aa386`: "The Helmsley Building ... Midtown Manhattan ..." | No |
| 4 | `p_658d6333`: "*Robert Von Ancken* is a prominent New York City real estate appraiser ..." | No |
| 5 | `p_e97ef7e6`: "Lenox Hill Neighborhood House ... East Side of Manhattan ..." | Yes |

**Answer synthesis.** After $\tau=5$ iterations, the canonicalizer $\kappa$ compacts the selected set $M_\tau$ (including `p_a73a8d8f` and the Von Ancken paragraph `p_658d6333`) into a provenance-preserving package. The head answers using only $(q, \kappa(M_\tau))$:

$$\hat{y} = \text{Art Deco-style skyscraper.}$$

The prediction matches the gold answer. Runtime profile: selection latency $\approx 32{,}153$ ms, head latency $\approx 838$ ms, total $\approx 41{,}487$ ms; iterations $= 5$.

**Takeaway.** The head's guidance steers the iterator directly to a paragraph that states both the location (East Side of Midtown) and the architectural style (Art Deco) of the relevant building (Chrysler Building), while additional picks provide neighborhood and appraiser context. Provenance in $\kappa(M_\tau)$ supports auditable linking from the final answer to its evidence.

A.6   STATEMENT FOR THE USE OF LARGE LANGUAGE MODELS (LLMs)

We used large language models (LLMs) as general-purpose tools for *writing assistance* and *engineering support*. For writing, we employed LLMs to improve clarity and style (e.g., rephrasing sentences, tightening paragraphs, standardizing notation, and proofreading grammar). Drafting, technical claims, algorithms, proofs, experiment design, and all final wording were authored and verified by the authors. For engineering, we consulted LLMs for debugging; all research code, data processing, and experiment scripts were implemented, audited, and executed by the authors. No text or code generated by an LLM was used verbatim without author review; we take full responsibility for the content.

