# OpenReview forum: "Hierarchical Sequence Iteration for Heterogeneous Question Answering"
_ICLR.cc/2026/Conference — ICLR 2026 Conference Withdrawn Submission_

### Official Review · Reviewer_Daog · 2025-10-29

**Soundness:** 2
**Presentation:** 1
**Contribution:** 2
**Rating:** 2
**Confidence:** 4

**Summary:**

This paper focuses on hybrid question answering systems involving heterogeneous information.
It aims to standardize retrieval information of different structures to enable unified utilization of heterogeneous data, and also improves the retrieval efficiency with multi-agent systems.

The paper proposes the Hierarchical Sequence (HSEQ) to address the above issues:
1. The authors convert heterogeneous information (e.g., plain text, tables, knowledge graphs) into a unified hierarchical sequence format, using identifiers to locate target information across different sources (e.g., rows and columns in tables), achieving a universal data representation.
2. A multi-agent collaborative framework is proposed for retrieval, with different agents responsible for global and local tasks respectively. This ensures sufficient retrieval evidence while avoiding resource waste.
3. The paper represents evidence in a structured manner, providing traceability and enhancing faithfulness during reasoning.

The authors conducted experiments on question-answering datasets with different structures, including pure text, tables, and knowledge graphs.

**Strengths:**

1. This paper focuses on leveraging heterogeneous retrieved information. Through an adapter approach, it unifies information of varying structures into a hierarchical sequence representation, enabling seamless integration of heterogeneous data.

2. The paper proposes a multi-agent collaborative framework, where different agents handle macro-level global planning and micro-level detail processing respectively. This improves retrieval accuracy while avoiding resource waste.

3. This paper conducts experiments on several heterogeneous datasets, including text, tables, and knowledge graphs, to validate its claims.

**Weaknesses:**

1. The author uses hierarchical sequences to represent heterogeneous unified information. However, the differences between representations of institutional information, such as plain text, tables, and knowledge graphs, are significant. Can this hierarchical sequence representation, serving as middleware, provide accurate guidance for subsequent retrieval? How much error is introduced during this transformation process, and how much retrieval noise does it cause? The article does not discuss the noise and information loss brought about by this intermediate transformation process.

2. The multi-agent-guided retrieval method proposed in the article is not novel. Using multi-agents for high-level planning and low-level specific operations is already a relatively common approach in the current field of llm agents, and its contribution is limited.

3. The writing of this article is somewhat disorganized, mainly reflected in the methodology section (Section 2).
    - The structure is messy and redundant: Core concepts (e.g., the unified sequence of HSEQ, the three-module design) are repeated across subsections (2.1, 2.2, 2.3), lacking logical coherence. This forces readers to piece together the framework themselves rather than following a linear and coherent explanation.
    - Key components (e.g., the fragment format of the HSEQ-Adapter, the "structure-aware neighborhood operator" of HSEQ-I) lack concrete examples or clear definitions—there are no sample fragments for tables/knowledge graphs, nor is it explained how iterative logic (e.g., parent-child jumps, window refreshing) operates in practice

**Questions:**

Please see weakness section

---

> ### Author Response · Authors · 2025-11-28
> **Response to Reviewer Daog**
>
> Thank you for the detailed and thoughtful review. Following reply focus on (i) potential noise / information loss from the hierarchical sequence representation, (ii) the originality of the multi-agent design, and (iii) clarity and redundancy in the methodology section, especially around HSEQ-Adapter and HSEQ-Iterator.
>
> > **R1. Information loss / noise in the hierarchical sequence representation**
>
> HSEQ structure can store both semantic and locational information , specifically:
>
> - For **text**, segments carry: parent chain of the sentence(last sentence, next sentence, location in article, etc,.), as well as the context
> - For **tables**, segments carry: table id and row/column identifiers, the header schema in metadata, and a serialized row or cell view.
> - For **KGs**, segments carry: the exact triple (h,r,t,[time])(h, r, t, [\text{time}])(h,r,t,[time]), stable node ids, and optional graph-level attributes (e.g., source KG, link info).
>
> Formally, the paper provides a *faithful linearization* theorem (Appendix, A2 to show the structure is probabilistic complete under stochastic selection, and every atomic support can be traced back to a unique original segment.
>
> Also from experiments:
>
> - On **HotpotQA** (text-only), HSEQ-based retrieval matches or exceeds text-only baselines, indicating that paragraph/sentence structure and discourse cues are preserved.
> - On **HybridQA** and **TAT-QA** (table + paragraph), the framework recovers the required row-level and cell-level evidence sufficiently to outperform or closely match specialized table–text models.
> - On **MetaQA** and **HotpotQA** (KG), HSEQ achieves strong performance on multi-hop reasoning, suggesting that multi-hop graph structure is preserved.
>
> > **R2. Novelty of the multi-agent guided retrieval design**
>
>
>
>  The review is correct that high-level planning + low-level execution is a common pattern in LLM agent systems. HSEQ does not claim conceptual novelty for “having two agents” per se. The key insight is that the unified interface and multi-agent control indeed build on themes explored in prior RAG and agentic systems. The contribution of HSEQ is not the existence of linearization or agents per se, but the specific combination of:
>
> 1. **Reversible, structure-aware hierarchical sequence across text, tables, and KGs.**Many prior works serialize heterogeneous data into flat text or task-specific graph formats. Also HSEQ is reversibale and traceable
> 2. **A unified iterator policy trained once and shared across modalities.** A single HSEQ-I is trained to operate over a common action space (select / expand / move window).
>
>
>
> Thinking from a perspective of data retrieval, we think we are the first one using this unique, revearsable and effective-retrievable framework.
>
> > **R3. Writing quality and redundancy in the methodology section**
>
>
>
>  This criticism is well taken. The original method section may be hard to follow the overall framework.
>
> We will imporve the writing with clearer modularization. Describing HSEQ from high-level architecture in Section 3 and adding more technique details in Section 4
>
>
> > **Summary**
>
> We thank the reviewer again for the helpful comments. The revised version will include modification to both section 3 and section 4, as well as adding some extra experiments, potentially suggested by reviewer qzgH, including another round of comparison with TAT-LLM[1], and potentially on CRAG[2]. We sincerely hope to hear from you about any extra advices or more questions.
>
>
>
> We hope comment above will solve your concerns.
>
>
>
> [1] Zhu, F., Liu, Z., Feng, F., Wang, C., Li, M., & Chua, T. S. (2024, November). Tat-llm: A specialized language model for discrete reasoning over financial tabular and textual data. In *Proceedings of the 5th ACM International Conference on AI in Finance* (pp. 310-318).
>
> [2] Yang, X., Sun, K., Xin, H., Sun, Y., Bhalla, N., Chen, X., ... & Dong, X. L. (2024). Crag-comprehensive rag benchmark. *Advances in Neural Information Processing Systems*, *37*, 10470-10490.

---

### Official Review · Reviewer_Mpqh · 2025-10-29

**Soundness:** 2
**Presentation:** 2
**Contribution:** 2
**Rating:** 2
**Confidence:** 4

**Summary:**

This paper investigates hybrid question answering over heterogeneous data sources. The main goal is to establish a unified way to handle information from multiple structures—such as text, tables, and knowledge graphs—while improving retrieval performance through a multi-agent system.
The authors introduce a Hierarchical Sequence (HSEQ) framework that converts heterogeneous information into a unified representation. Each piece of data, regardless of its original form, is encoded as a hierarchical sequence with explicit identifiers (e.g., for table cells or graph nodes), enabling consistent access across sources.
To improve retrieval efficiency, a multi-agent coordination mechanism is adopted: different agents are responsible for global and local retrieval tasks. This setup aims to balance retrieval thoroughness with computational efficiency. The authors further claim that this structure-aware representation enhances reasoning faithfulness and allows traceable evidence construction.
Experiments are performed on question-answering benchmarks covering multiple modalities, including textual, tabular, and knowledge-graph-based datasets.

**Strengths:**

The paper tackles an important problem—how to effectively retrieve and integrate information from heterogeneous sources for QA—by putting forward an interesting idea of using a hierarchical unified representation for multiple data types (which could simplify downstream reasoning), proposing a multi-agent retrieval strategy that provides a modular way to combine global exploration with local evidence refinement, and evaluating its approach on diverse datasets covering different structural formats to offer a broad view of its applicability，and this approach aligns with the framework introduced in the paper that unifies text, tables, and knowledge graphs into a reversible hierarchical sequence and uses a Head Agent and an Iteration Agent for guided, budget-aware retrieval and reasoning.

**Weaknesses:**

The proposed hierarchical sequence representation may oversimplify structural differences among text, tables, and knowledge graphs. It is unclear whether this abstraction preserves sufficient semantics for accurate retrieval. The paper lacks analysis on information loss or noise introduced by this transformation.

The writing in the methodology section of this paper is somewhat confusing. Could the authors clarify how each module of the proposed method interacts and operates? Providing some examples would be helpful, as the current presentation makes comprehension difficult.

The overall novelty of this paper is relatively limited. Neither the unified retrieval intermediary nor the multi-agent collaborative RAG workflow demonstrates significant methodological novelty or offers new insights.

**Questions:**

I have no further questions.

---

> ### Author Response · Authors · 2025-11-28
> **Response to Reviewer Mpqh**
>
> Thank you for the careful reading. Our reply will mainly touch on three central aspects: (i) whether the hierarchical sequence preserves enough structure/semantics, (ii) clarity of the methodology and module interactions, and (iii) the novelty of the unified interface and multi-agent design. Each is addressed in turn.
>
> > R1. Does the hierarchical sequence oversimplify structure and lose semantics?
>
>
>
>  HSEQ structure can store both semantic and locational information , specifically:
>
> - For **text**, segments carry: parent chain of the sentence(last sentence, next sentence, location in article, etc,.), as well as the context
> - For **tables**, segments carry: table id and row/column identifiers, the header schema in metadata, and a serialized row or cell view.
> - For **KGs**, segments carry: the exact triple (h,r,t,[time])(h, r, t, [\text{time}])(h,r,t,[time]), stable node ids, and optional graph-level attributes (e.g., source KG, link info).
>
> Formally, the paper provides a *faithful linearization* theorem (Appendix, A2 to show the structure is probabilistic complete under stochastic selection, and every atomic support can be traced back to a unique original segment.
>
> Also from experiments:
>
> - On **HotpotQA** (text-only), HSEQ-based retrieval matches or exceeds text-only baselines, indicating that paragraph/sentence structure and discourse cues are preserved.
> - On **HybridQA** and **TAT-QA** (table + paragraph), the framework recovers the required row-level and cell-level evidence sufficiently to outperform or closely match specialized table–text models.
> - On **MetaQA** and **HotpotQA** (KG), HSEQ achieves strong performance on multi-hop reasoning, suggesting that multi-hop graph structure is preserved.
>
>
>
> > R2. Clarity of methodology and module interactions (HSEQ-A / HSEQ-I / HSEQ-H)
>
>
>
>  This is very helpful feedback. We will imporve the writing with clearer modularization. Describing HSEQ from high-level architecture in Section 3 and adding more technique details in Section 4
>
>
>
> Additionally, the appendix now includes **step-by-step case studies** with actual questions under HybridQA and HotpotQA. We copy one part directly from in Section A5.2:
>
>
>
> #### **Setup**
>
> Query $q$: *“Who is the author of the novel that inspired the 2004 Russian film directed by Timur Bekmambetov?”*
> HSEQ-I (iterator): `Qwen3-4B-Instruct-2507`
> HSEQ-H (head): `Falcon-H1-7B-Instruct`
> Guidance mode: `head`; source: cache (latency ≈ 0.12 ms).
>
> #### **Head-generated guidance**
>
> The head planner emits a short plan:
> (i) identify the 2004 Russian film directed by Bekmambetov;
> (ii) locate the novel that inspired it;
> (iii) stop once the *author of that novel* is found.
> This plan is injected as a prefix and acts as a soft prior on where the iterator should probe first.
>
> #### **Guided iteration over $S_h$**
>
> The iterator consumes the guidance and operates over the HSEQ stream with a fixed window and top-$k$ selection.
> Table 1 summarizes the six steps (all sufficiency flags were `false`; the loop terminates by budget).
>
> ##### **Table 1. Stepwise selection (abridged)**
>
> Segment ids prefixed by level: `p_` (paragraph), `row_` (table row).
>
> | **Step** | **Key picks (content excerpt)**                              | **Sufficient?** |
> | -------- | ------------------------------------------------------------ | --------------- |
> | **1**    | `p_6df9c849`: “*Night Watch* (…) is a 2004 Russian … directed by Timur Bekmambetov. It is loosely based on the novel *The Night Watch* by Serg[e{i} Lukyanenko]…” | No              |
> | **2**    | `p_c15173df`, `p_3bc4a108`, `p_54f6ef94`: contextual paragraphs (“List of Russian films of 2004”, “2004” entries) | No              |
> | **3**    | `row_a44a4a17`: table row confirming *Night Watch* with director “Timur Bekmambetov” | No              |
> | **4–6**  | additional table rows from the same list (*Arie*, *Countdown*, *Dad or Papa*, etc.) providing film set context | Yes             |
>
> #### **Answer synthesis**
>
> After $\tau = 6$ iterations, the canonicalizer $\kappa$ compacts the selected set $M_\tau$ (paragraph + corroborating table rows) into a provenance-preserving package (segment ids, levels, offsets, snippets).
> The head $\mathcal{H}$ is prompted *only* with $(q, \kappa(M_\tau))$ and outputs:
>
> \[
> \hat{y} = \text{Sergei Lukyanenko}.
> \]
>
> The prediction matches the gold answer (EM/F1 = 1.0).
> Runtime profile: selection latency ≈ 32,185 ms, head latency ≈ 1,826 ms, total ≈ 34,011 ms; number of iterations = 6.

---

> ### Author Response · Authors · 2025-11-28
> **Continue of Response to Reviewer Mpqh**
>
> > R3. Novelty of unified interface and multi-agent collaborative RAG
>
>
>
>  The unified interface and multi-agent control indeed build on themes explored in prior RAG and agentic systems. The contribution of HSEQ is not the existence of linearization or agents per se, but the specific combination of:
>
> 1. **Reversible, structure-aware hierarchical sequence across text, tables, and KGs.**Many prior works serialize heterogeneous data into flat text or task-specific graph formats. Also HSEQ is reversibale and traceable
> 2. **A unified iterator policy trained once and shared across modalities.** A single HSEQ-I is trained to operate over a common action space (select / expand / move window).
>
>
>
> Thinking from a perspective of data retrieval, we think we are the first one using this unique, revearsable and effective-retrievable framework.
>
> > Summary
>
> We thank the reviewer again for the helpful comments. The revised version will include modification to
> 1) both section 3 and section 4, to facilitate understanding of the overall framework
> 2) adding some extra experiments, potentially suggested by reviewer qzgH, including another round of comparison with TAT-LLM[1], and potentially on CRAG[2]. We sincerely hope to hear from you about any extra advices or more questions.
>
> We hope comment above will solve your concerns.
>
>
> [1] Zhu, F., Liu, Z., Feng, F., Wang, C., Li, M., & Chua, T. S. (2024, November). Tat-llm: A specialized language model for discrete reasoning over financial tabular and textual data. In *Proceedings of the 5th ACM International Conference on AI in Finance* (pp. 310-318).
>
> [2] Yang, X., Sun, K., Xin, H., Sun, Y., Bhalla, N., Chen, X., ... & Dong, X. L. (2024). Crag-comprehensive rag benchmark. *Advances in Neural Information Processing Systems*, *37*, 10470-10490.

---

### Official Review · Reviewer_qzgH · 2025-10-31

**Soundness:** 2
**Presentation:** 2
**Contribution:** 2
**Rating:** 2
**Confidence:** 3

**Summary:**

The paper introduces a unified framework for question answering (QA) that linearizes heterogeneous knowledge sources (text, tables, and knowledge graphs (KGs)) into a single, reversible interface. It enables structure-aware, budgeted evidence collection before synthesizing an answer. Experiments on multiple benchmarks demonstrate improved performance over several baselines.

**Strengths:**

* The paper is well structured.
* Ablation studies are reported, demonstrating the contribution of each component in the proposed framework.

**Weaknesses:**

* Unfair comparison with prior work: The paper does not employ the same base model when comparing with the existing works. For example, the TAT-LLM baseline [1] in Table 2 is based on Llama 2, while the closest base model used in this paper is Llama 3.1, and the base models of the proposed framework in Table 2 are not from the Llama series. This makes it difficult to attribute performance gains solely to the proposed framework rather than differences in underlying model capacity.
* Limited coverage of heterogeneous QA challenges in the chosen datasets: The experimental evaluation does not fully capture the diversity of heterogeneous QA. In particular, MetaQA is the only benchmark involving KGs, and it does not combine KGs with other knowledge sources. Incorporating more comprehensive benchmarks (e.g., CRAG [2]) could strengthen the motivation and empirical justification for the proposed approach.
* Lack of clarity in experimental details and results: Please see the questions below.

[1] TAT-LLM: A Specialized Language Model for Discrete Reasoning over Tabular and Textual Data.

[2] CRAG -- Comprehensive RAG Benchmark.

**Questions:**

* Why the results in Table 2 and Table 3 are inconsistent (e.g., the F1 of HybridQA for HSEQ (best))?
* Why some metrics are not reported (i.e., those dashes) in Table 2 and Table 5?

---

> ### Author Response · Authors · 2025-11-28
> **Response to Reviewer qzgH**
>
> Thank you for the detailed review. Below is a point-by-point response addressing the concerns regarding fairness, dataset coverage, and clarity of results.
>
>
>
> > **R1. Fairness of comparison with prior models (LLM series mismatch)**
>
>
>  Although different benchmark do give different result, we still think HSEQ outperform those baselines, apart from extra experiment on Llama 2 series model, our HSEQ result shown in Table 2 only finetuned a 4B model (Qwen3-4B-Instruct-2507) and an un-finetuned 7B model for reasoning, while TAT-LLM, although indeed used somewhat 'old' model, are still at least finetuning a 7B one.
>
>
>
> We are adding more experiments, but we are not 'comparing unfairly'. We will offer more results soon.
>
>
> Also, **HSEQ uses multiple LLMs across families (Falcon, Llama-3.1, Qwen-3B/4B, DeepSeek)** to show that performance gains are *consistent across heterogeneous backbone models*, rather than requiring series-matched models.
>
>
>
> > **R2. Dataset coverage and limited diversity of heterogeneous QA**
>
> MetaQA is actually not the only one using KG, HotpotQA also has KG entries. And we compared our result with HippoRAG, GcR, ToG, AdaptiveRAG on HotpotQA, HSEQ outperform them.
>
>
>
> > **R3. Inconsistency between Table 2 and Table 3**
>
>
>
>  There might be some misunderstanding, but **in Table 2 we listed HSEQ's best and median result, while in Table 3 we just show the average results**.  We will add more notes on the table for this, thanks for raising this point.
>
> > **R4. Missing metrics (dashes in Table 2 and Table 5)**
>
> In table 2, Shaded cells (N/A) indicate the method is not applicable to that benchmark; gray dashes (–) indicate metric not reported. Although mentioned in the table title, it may be ambiguoue, we will add extra footnote indicates that this metric is not provided by the official benchmark or baseline implementation. Also, for LLM-only QA usage, we didn't consider F1 as it's zero-shot prompting result, we think accuracy is enough to show those methods are not as good as RAG-based methods. Hope this removes ambiguity.
>
> > **Summary**
>
> We thank the reviewer again for the helpful comments. The revised version will include modification to
> 1) both section 3 and section 4, to facilitate understanding of the overall framework
> 2) adding some extra experiments, including another round of comparison with TAT-LLM[1], and potentially on CRAG[2]. We sincerely hope to hear from you about any extra advices or more questions.
>
> We hope comment above will solve your concerns.
>
>
> [1] Zhu, F., Liu, Z., Feng, F., Wang, C., Li, M., & Chua, T. S. (2024, November). Tat-llm: A specialized language model for discrete reasoning over financial tabular and textual data. In *Proceedings of the 5th ACM International Conference on AI in Finance* (pp. 310-318).
>
> [2] Yang, X., Sun, K., Xin, H., Sun, Y., Bhalla, N., Chen, X., ... & Dong, X. L. (2024). Crag-comprehensive rag benchmark. *Advances in Neural Information Processing Systems*, *37*, 10470-10490.

---

### Official Review · Reviewer_Z5RG · 2025-10-31

**Soundness:** 2
**Presentation:** 2
**Contribution:** 1
**Rating:** 4
**Confidence:** 3

**Summary:**

The paper introduces Hierarchical Sequence (HSEQ) Iteration, a unified, format-agnostic framework for multi-step, multi-source QA. It linearizes heterogeneous sources into a hierarchical sequence and runs budget-aware iterative retrieval with a planning/answering head. Results show solid accuracy and improved efficiency on several multi-hop QA benchmarks.

**Strengths:**

1. Unified interface across text, tables, and KG. Consistent control and auditing.

2. Competitive or SOTA accuracy on diverse multi-hop datasets.

3. Clear efficiency gains vs. graph-heavy pipelines on comparable quality.

**Weaknesses:**

1. **Complexity/Cost issue.** The iteration processing increases the complexity and cost of learning.

2. **The unified interface, which is the core contribution of this paper, lacks novelty.** Positioning vs. prior unified/structural RAG is not fully convincing. The authors should clarify what is technically new and why it matters.

3. Lack of ablation studies on the multi-agent design.

**Questions:**

1. Could the author analyze the efficiency trade-off between LLM-only and Graph ToG?

---

> ### Author Response · Authors · 2025-11-28
> **Response to Reviewer Z5RG**
>
> We thank the reviewer for the constructive feedback and for highlighting both the strengths (unified interface, efficiency, strong accuracy) and the main concerns (complexity, novelty, and ablations). Here are our reply with potential revisions
>
>
> > **Complexity / cost of iterative processing**
>
>
>
>  HSEQ is designed to limit complexity during inference-time by:
>
> - The adapter HSEQ-A runs once per corpus to build the hierarchical sequence `S_h`. This replaces online graph traversal and dynamic subgraph construction used in graph-heavy pipelines like ToG.
> - Itertor HSEQ-I is effectively dealing with sequence and the iteration (typically 4–6 steps), tokens and steps will be saved compared with prompt-only method.
> - Retrieval complexity is controlled by small fixed knobs `(W, k, T_max)`, so inference scales with *evidence actually inspected*, not corpus size. In practice, HSEQ runs with 3–5 steps, while ToG may expand large neighborhoods proportional to graph size.
>
> > **Novelty of the unified interface vs. prior unified/structural RAG**
>
> HSEQ does not merely linearize structure; it introduces a *reversible, provenance-aware hierarchical sequence* specifically designed to support budgeted iterative retrieval with explicit stopping. The key new elements are:
>
> 1. Reversible, provenance-keeping hierarchical sequence across text, tables, and KG.
>     Each segment stores:
>    - level (sentence/paragraph/row/cell/triple),
>    - parent pointer,
>    - alignment metadata (offsets/coordinates),
>    - stable IDs.
>       This guarantees reconstructability and exact provenance, which prior “unified” formats typically do not preserve.
> 2. A single learned iteration policy across modalities with an explicit sufficiency head.
>     The same policy and action space operate for text, tables, and KG. Prior unified RAG methods usually require separate retrievers, separate graph agents, or lack a learned stopping signal.
> 3. Decoupled adapter / iterator / head with a canonical evidence boundary.
>     The canonicalizer ensures the head always receives the same structured evidence package, enabling:
>    - plug-and-play swapping of iterator/head LLMs,
>    - clean provenance,
>    - auditable and refinement-capable answering.
>       This explicit modularity is absent in most structural RAG systems.
> 4. Theoretical properties:
>     The appendix includes proofs of reconstructability, bounded selection complexity, and halting guarantees—properties dependent on the exact HSEQ design and not covered in prior work.
>
>
>
>  We will strengthen Related Work to group prior structural RAG references and explicitly clarify how HSEQ’s interface and guarantees differ.
>
> > **Ablations on the multi-agent design**
>
>
>
>  Several existing ablations already correspond directly to the multi-agent components:
>
> - LLM-only --> removes *both* HSEQ-A and HSEQ-I.
> - No SFT--> isolates the impact of the learned iteration policy.
> - No guidance--> removes head-generated or heuristic plan.
> - Heuristic-only guidance  -->isolates *learned vs. heuristic* guidance.
>
> These directly map to turning on/off HSEQ-A, HSEQ-I, and the guidance role of HSEQ-H.
>
>
> > **Efficiency trade-off: LLM-only vs. Graph-ToG**
>
>
>  This comparison between LLM only and Graph-ToG aligns with HSEQ’s motivation. Conceptually:
>
> - **LLM-only**
>   - Pros: single call, lowest latency.
>   - Cons: poor multi-hop performance; no controlled evidence retrieval.
> - **Graph ToG**
>   - Pros: strong structural reasoning on KG.
>   - Cons: high latency due to large neighborhood expansions and repeated LLM calls.
> - **HSEQ**
>   - Middle ground: bounded steps and windowed segments, which is stable, predictable latency.
>   - Accuracy consistently matches or exceeds ToG, without graph-expansion overhead.
>
>
>
> > **Summary**
>
> We thank the reviewer again for the helpful comments. The revised version will include modification to
> 1) both section 3 and section 4, to facilitate understanding of the overall framework
> 2) adding some extra experiments, potentially suggested by reviewer qzgH, including another round of comparison with TAT-LLM[1], and potentially on CRAG[2]. We sincerely hope to hear from you about any extra advices or more questions.
>
> We hope comment above will solve your concerns.
>
> [1] Zhu, F., Liu, Z., Feng, F., Wang, C., Li, M., & Chua, T. S. (2024, November). Tat-llm: A specialized language model for discrete reasoning over financial tabular and textual data. In *Proceedings of the 5th ACM International Conference on AI in Finance* (pp. 310-318).
>
> [2] Yang, X., Sun, K., Xin, H., Sun, Y., Bhalla, N., Chen, X., ... & Dong, X. L. (2024). Crag-comprehensive rag benchmark. *Advances in Neural Information Processing Systems*, *37*, 10470-10490.

---

### Author Response · Authors · 2025-12-04
**Summary for the rebuttal**

We appreciate the reviewers’ detailed feedback and have substantially revised the paper to address the main concerns around clarity, novelty positioning, and empirical validation. Below is a brief summary of the changes and how they respond to specific points.

> **1. Clarifying the technical novelty of HSEQ vs. prior unified / structural RAG**

To address the concern that the unified interface “lacks novelty” and is not clearly positioned versus structural RAG:

We added a dedicated paragraph **“Structural and unified RAG interfaces”** in the Related Work section, grouping prior structural / graph-based / hierarchical RAG systems and explicitly contrasting them with HSEQ.

Our HSEQ has three technical differences:

1. **Reversible, modality-preserving representation:** HSEQ keeps modality-specific structure (text / table / KG) and alignment (offsets, coordinates, IDs) in the segment schema, instead of collapsing everything into a single opaque graph index.
2. **Segment schema as first-class interface:** The unified segment format (id, level, parent, content, metadata) is used as the *only* interface to the controller, enabling structure-aware navigation and sufficiency-based stopping, rather than merely adding structure for re-ranking or expansion.
3. **Formal properties and sufficiency-based control:** We provide a precise definition of faithful linearization and budgeted selection, plus a sufficiency-based iteration loop with guaranteed halting under fixed budgets, which is not present in prior structural RAG work.

- These distinctions are now clearly articulated in Related Work and referenced in the Method and Properties discussions.

> **2. Improving clarity and structure of the methodology**

Section 4, Using HSEQ on LLMs are rewritten.

> **3. Efficiency trade-off: LLM-only vs. ToG vs. HSEQ**

Reviewers requested analysis of the efficiency trade-off between LLM-only, graph-based methods (e.g., ToG), and HSEQ. We added analysis

> **4. Fairer comparison on TAT-QA and baseline consistency**

One reviewer pointed out that TAT-LLM in our original Table 2 was reported with a different base model (LLaMA 2) compared to our HSEQ heads:

- We re-ran **TAT-LLM** using **Qwen3-4B-Instruct-2507** as the base model to better match the capacity of our open-source backbones.
- The updated **TAT-QA** numbers for the re-run are:
  - **Acc = 73.1, F1 = 81.0** for TAT-LLM (Qwen3-4B).
- Tables and text have been updated to reflect these new numbers, and we now clearly state which backbone is used for each baseline.
- HSEQ still outperforms this stronger TAT-LLM baseline variant on TAT-QA, but now under a more comparable model setting.



> **5. Example added on HSEQ-A**

Examples are added on HSEQ-A to support the design choices.



Overall, the revision aims to:

1. Clarify and sharpen the technical contribution of HSEQ relative to structural/unified RAG,
2. Improve the readability and coherence of the method section,
3. Provide a more careful and fair empirical comparison (including the re-run of TAT-LLM under a comparable backbone),
4. Explicitly analyze efficiency–accuracy trade-offs among LLM-only, ToG, and HSEQ, and
5. Strengthen examples to support the design choices.

We hope these changes address the reviewers’ concerns and help the AC assess the paper more favorably.

---

### Note · Authors · 2026-01-05

I have read and agree with the venue's withdrawal policy on behalf of myself and my co-authors.